# Hierarchical Clustering via Spreading Metrics

Aurko Roy[1] and Sebastian Pokutta[2]

[1]College of Computing, Georgia Institute of Technology, Atlanta, GA, USA.
*Email:* aurko@gatech.edu
[2]ISyE, Georgia Institute of Technology, Atlanta, GA, USA.
*Email:* sebastian.pokutta@isye.gatech.edu

## Abstract

We study the cost function for hierarchical clusterings introduced by [16] where hierarchies are treated as first-class objects rather than deriving their cost from projections into flat clusters. It was also shown in [16] that a top-down algorithm returns a hierarchical clustering of cost at most $O\left(\alpha_n \log n\right)$ times the cost of the optimal hierarchical clustering, where $\alpha_n$ is the approximation ratio of the Sparsest Cut subroutine used. Thus using the best known approximation algorithm for Sparsest Cut due to Arora-Rao-Vazirani, the top-down algorithm returns a hierarchical clustering of cost at most $O\left(\log^{3/2} n\right)$ times the cost of the optimal solution. We improve this by giving an $O(\log n)$-approximation algorithm for this problem. Our main technical ingredients are a combinatorial characterization of ultrametrics induced by this cost function, deriving an Integer Linear Programming (ILP) formulation for this family of ultrametrics, and showing how to iteratively round an LP relaxation of this formulation by using the idea of *sphere growing* which has been extensively used in the context of graph partitioning. We also prove that our algorithm returns an $O(\log n)$-approximate hierarchical clustering for a generalization of this cost function also studied in [16]. We also give constant factor inapproximability results for this problem.

## 1 Introduction

*Hierarchical clustering* is an important method in cluster analysis where a data set is recursively partitioned into clusters of successively smaller size. They are typically represented by rooted trees where the root corresponds to the entire data set, the leaves correspond to individual data points and the intermediate nodes correspond to a cluster of its descendant leaves. Such a hierarchy represents several possible *flat clusterings* of the data at various levels of granularity; indeed every pruning of this tree returns a possible clustering. Therefore in situations where the number of desired clusters is not known beforehand, a hierarchical clustering scheme is often preferred to flat clustering.

The most popular algorithms for hierarchical clustering are bottoms-up agglomerative algorithms like *single linkage*, *average linkage* and *complete linkage*. In terms of theoretical guarantees these algorithms are known to correctly recover a ground truth clustering if the similarity function on the data satisfies corresponding stability properties (see, e.g., [5]). Often, however, one wishes to think of a good clustering as optimizing some kind of cost function rather than recovering a hidden "ground truth". This is the standard approach in the classical clustering setting where popular objectives are $k$-means, $k$-median, min-sum and $k$-center (see Chapter 14, [23]). However as pointed out by [16] for a lot of popular hierarchical clustering algorithms including linkage based algorithms, it is hard to pinpoint explicitly the cost function that these algorithms are optimizing. Moreover, much of the existing cost function based approaches towards hierarchical clustering evaluate a hierarchy based

on a cost function for flat clustering, e.g., assigning the $k$-means or $k$-median cost to a pruning of this tree. Motivated by this, [16] introduced a cost function for hierarchical clustering where the cost takes into account the entire structure of the tree rather than just the projections into flat clusterings. This cost function is shown to recover the intuitively correct hierarchies on several synthetic examples like planted partitions and cliques. In addition, a top-down graph partitioning algorithm is presented that outputs a tree with cost at most $O(\alpha_n \log n)$ times the cost of the optimal tree and where $\alpha_n$ is the approximation guarantee of the Sparsest Cut subroutine used. Thus using the Leighton-Rao algorithm [33] or the Arora-Rao-Vazirani algorithm [3] gives an approximation factor of $O\left(\log^2 n\right)$ and $O\left(\log^{3/2} n\right)$ respectively.

In this work we give a polynomial time algorithm to recover a hierarchical clustering of cost at most $O(\log n)$ times the cost of the optimal clustering according to this cost function. We also analyze a generalization of this cost function studied by [16] and show that our algorithm still returns an $O(\log n)$ approximate clustering in this setting. We do this by giving a combinatorial characterization of the ultrametrics induced by this cost function, writing a convex relaxation for it and showing how to iteratively round a fractional solution into an integral one using a rounding scheme used in graph partitioning algorithms. We also implement the integer program, its LP relaxation, and the rounding algorithm and test it on some synthetic and real world data sets to compare the cost of the rounded solutions to the true optimum, as well as to compare its performance to other hierarchical clustering algorithms used in practice. Our experiments suggest that the hierarchies found by this algorithm are often better than the ones found by linkage based algorithms as well as the $k$-means algorithm in terms of the error of the best pruning of the tree compared to the ground truth. We conclude with constant factor hardness results for this problem.

## 1.1 Related Work

The immediate precursor to this work is [16] where the cost function for evaluating a hierarchical clustering was introduced. Prior to this there has been a long line of research on hierarchical clustering in the context of phylogenetics and taxonomy (see, e.g., [22]). Several authors have also given theoretical justifications for the success of the popular linkage based algorithms for hierarchical clustering (see, e.g. [1]). In terms of cost functions, one approach has been to evaluate a hierarchy in terms of the $k$-means or $k$-median cost that it induces (see [17]). The cost function and the top-down algorithm in [16] can also be seen as a theoretical justification for several graph partitioning heuristics that are used in practice.

LP relaxations for hierarchical clustering have also been studied in [2] where the objective is to fit a tree metric to a data set given pairwise dissimilarities. Another work that is indirectly related to our approach is [18] where an ILP was studied in the context of obtaining the closest ultrametric to arbitrary functions on a discrete set. Our approach is to give a combinatorial characterization of the ultrametrics induced by the cost function of [16] which allows us to use the tools from [18] to model the problem as an ILP. The natural LP relaxation of this ILP turns out to be closely related to LP relaxations considered before for several graph partitioning problems (see, e.g., [33, 19, 32]) and we use a rounding technique studied in this context to round this LP relaxation.

Recently, we became aware of independent work by Charikar and Chatziafratis [12] obtaining similar results for hierarchical clustering. In particular they improve the approximation factor to $O\left(\sqrt{\log n}\right)$ by showing how to round a spreading metric SDP relaxation for this cost function. They also analyze a similar LP relaxation using the *divide-and-conquer approximation algorithms using spreading metrics* paradigm of [20] together with a result of [7] to prove an $O(\log n)$ approximation. Finally, they also give similar inapproximability results for this problem.

## 2 Preliminaries

A similarity based clustering problem consists of a dataset $V$ of $n$ points and a *similarity function* $\kappa : V \times V \to \mathbb{R}$ such that $\kappa(i, j)$ is a measure of the similarity between $i$ and $j$ for any $i, j \in V$. We will assume that the similarity function is symmetric, i.e., $\kappa(i, j) = \kappa(j, i)$ for every $i, j \in V$. We also require $\kappa \geq 0$ as in [16]; see supplementary material for a discussion. Note that we do not make any assumptions about the points in $V$ coming from an underlying metric space. For a given instance of a clustering problem we have an associated weighted complete graph $K_n$ with vertex set $V$ and

weight function given by $\kappa$. A *hierarchical clustering* of $V$ is a tree $T$ with a designated root $r$ and with the elements of $V$ as its leaves, i.e., $\text{leaves}(T) = V$. For any set $S \subseteq V$ we denote the *lowest common ancestor* of $S$ in $T$ by $\text{lca}(S)$. For pairs of points $i, j \in V$ we will abuse the notation for the sake of simplicity and denote $\text{lca}(\{i, j\})$ simply by $\text{lca}(i, j)$. For a node $v$ of $T$ we denote the subtree of $T$ rooted at $v$ by $T[v]$. The following cost function was introduced by [16] to measure the quality of the hierarchical clustering $T$

$$\text{cost}(T) := \sum_{\{i,j\} \in E(K_n)} \kappa(i, j) \left| \text{leaves}(T[\text{lca}(i, j)]) \right|. \tag{1}$$

The intuition behind this cost function is as follows. Let $T$ be a hierarchical clustering with designated root $r$ so that $r$ represents the whole data set $V$. Since $\text{leaves}(T) = V$, every internal node $v \in T$ represents a cluster of its descendant leaves, with the leaves themselves representing singleton clusters of $V$. Starting from $r$ and going down the tree, every distinct pair of points $i, j \in V$ will be eventually separated at the leaves. If $\kappa(i, j)$ is large, i.e., $i$ and $j$ are very similar to each other then we would like them to be separated as far down the tree as possible if $T$ is a good clustering of $V$. This is enforced in the cost function (1): if $\kappa(i, j)$ is large then the number of leaves of $\text{lca}(i, j)$ should be small, i.e., $\text{lca}(i, j)$ should be far from the root $r$ of $T$.

Under the cost function (1), one can interpret the tree $T$ as inducing an ultrametric $d_T$ on $V$ given by $d_T(i, j) := \left| \text{leaves}(T[\text{lca}(i, j)]) \right| - 1$. This is an ultrametric since $d_T(i, j) = 0$ iff $i = j$ and for any triple $i, j, k \in V$ we have $d_T(i, j) \leq \max\{d_T(i, k), d_T(j, k)\}$. The following definition introduces the notion of *non-trivial ultrametrics*. These turn out to be precisely the ultrametrics that are induced by tree decompositions of $V$ corresponding to cost function (1), as we will show in Lemma 5.

**Definition 1.** *An ultrametric $d$ on a set of points $V$ is* non-trivial *if the following conditions hold.*

1. *For every non-empty set $S \subseteq V$, there is a pair of points $i, j \in S$ such that $d(i, j) \geq |S| - 1$.*

2. *For any $t$ if $S_t$ is an equivalence class of $V$ under the relation $i \sim j$ iff $d(i, j) \leq t$, then $\max_{i,j \in S_t} d(i, j) \leq |S_t| - 1$.*

Note that for an equivalence class $S_t$ where $d(i, j) \leq t$ for every $i, j \in S_t$ it follows from Condition 1 that $t \geq |S_t| - 1$. Thus in the case when $t = |S_t| - 1$ the two conditions imply that the maximum distance between any two points in $S$ is $t$ and that there is a pair $i, j \in S$ for which this maximum is attained. The following lemma shows that non-trivial ultrametrics behave well under restrictions to equivalence classes $S_t$ of the form $i \sim j$ iff $d(i, j) \leq t$. Due to page limitation full proofs are included in the supplementary material.

**Lemma 2.** *Let $d$ be a non-trivial ultrametric on $V$ and let $S_t \subseteq V$ be an equivalence class under the relation $i \sim j$ iff $d(i, j) \leq t$. Then $d$ restricted to $S_t$ is a non-trivial ultrametric on $S_t$.*

The intuition behind the two conditions in Definition 1 is as follows. Condition 1 imposes a certain lower bound by ruling out trivial ultrametrics where, e.g., $d(i, j) = 1$ for every distinct pair $i, j \in V$. On the other hand Condition 2 discretizes and imposes an upper bound on $d$ by restricting its range to the set $\{0, 1, \ldots, n - 1\}$ (see Lemma 3). This rules out the other spectrum of triviality where for example $d(i, j) = n$ for every distinct pair $i, j \in V$ with $|V| = n$.

**Lemma 3.** *Let $d$ be a non-trivial ultrametric on the set $V$. Then the range of $d$ is contained in the set $\{0, 1, \ldots, n - 1\}$ with $|V| = n$.*

## 3 Ultrametrics and Hierarchical Clusterings

In this section we study the combinatorial properties of the ultrametrics induced by cost function (1). We start with the following easy lemma showing that if a subset $S \subseteq V$ has $r$ as its lowest common ancestor, then there must be a pair of points $i, j \in S$ for which $r = \text{lca}(i, j)$.

**Lemma 4.** *Let $S \subseteq V$ of size $\geq 2$. If $r = \text{lca}(S)$ then there is a pair $i, j \in S$ such that $\text{lca}(i, j) = r$.*

The following lemma shows that non-trivial ultrametrics exactly capture the ultrametrics that are induced by tree decompositions of $V$ using cost function (1). The proof of Lemma 5 is inductive and uses Lemma 4 as a base case. As it turns out, the inductive proof also gives an algorithm to build the corresponding hierarchical clustering given such a non-trivial ultrametric in polynomial time. Since

this algorithm is relatively straightforward, we refer the reader to the supplementary material for the details.

**Lemma 5.** *Let $T$ be a hierarchical clustering on $V$ and let $d_T$ be the ultrametric on $V$ induced by cost function (1). Then $d_T$ is a non-trivial ultrametric on $V$. Conversely, let $d$ be a non-trivial ultrametric on $V$. Then there is a hierarchical clustering $T$ on $V$ such that for any pair $i, j \in V$ we have $d_T(i, j) = |\text{leaves}(T[\text{lca}\,(i,j)])| - 1 = d(i,j)$. Moreover this hierarchy can be constructed in time $O\left(n^3\right)$ where $|V| = n$.*

Therefore to find the hierarchical clustering of minimum cost, it suffices to minimize $\langle \kappa, d \rangle$ over non-trivial ultrametrics $d : V \times V \to \{0, \ldots, n-1\}$. A natural approach is to formulate this problem as an Integer Linear Program (ILP) and then study Linear Programming (LP) relaxations of it. We consider the following ILP for this problem that is motivated by [18]. We have the variables $x_{ij}^1, \ldots, x_{ij}^{n-1}$ for every distinct pair $i, j \in V$ with $x_{ij}^t = 1$ if and only if $d(i, j) \geq t$. For any positive integer $n$, let $[n] := \{1, 2, \ldots, n\}$.

$$\min \quad \sum_{t=1}^{n-1} \sum_{\{i,j\} \in E(K_n)} \kappa(i,j) x_{ij}^t \qquad \text{(ILP-ultrametric)}$$

$$\text{s.t.} \quad x_{ij}^t \geq x_{ij}^{t+1} \qquad \forall i, j \in V, t \in [n-2] \qquad (2)$$

$$x_{ij}^t + x_{jk}^t \geq x_{ik}^t \qquad \forall i, j, k \in V, t \in [n-1] \qquad (3)$$

$$\sum_{i,j \in S} x_{ij}^t \geq 2 \qquad \forall t \in [n-1], S \subseteq V, |S| = t+1 \qquad (4)$$

$$\sum_{i,j \in S} x_{ij}^{|S|} \leq |S|^2 \left( \sum_{i,j \in S} x_{ij}^t + \sum_{\substack{i \in S \\ j \notin S}} \left(1 - x_{ij}^t\right) \right) \forall t \in [n-1], S \subseteq V \qquad (5)$$

$$x_{ij}^t = x_{ji}^t, x_{ii}^t = 0 \qquad \forall i, j \in V, t \in [n-1] \qquad (6)$$

$$x_{ij}^t \in \{0, 1\} \qquad \forall i, j \in V, t \in [n-1] \qquad (7)$$

Note that constraint (3) is the same as the *strong triangle inequality* since the variables $x_{ij}^t$ are in $\{0, 1\}$. Constraint 6 ensures that the ultrametric is symmetric. Constraint 4 ensures the ultrametric satisfies Condition 1 of non-triviality: for every $S \subseteq V$ of size $t+1$ we know that there must be points $i, j \in S$ such that $d(i, j) = d(j, i) \geq t$ or in other words $x_{ij}^t = x_{ji}^t = 1$. Constraint 5 ensures that the ultrametric satisfies Condition 2 of non-triviality. To see this note that the constraint is active only when $\sum_{i,j \in S} x_{ij}^t = 0$ and $\sum_{i \in S, j \notin S}(1 - x_{ij}^t) = 0$. In other words $d(i, j) \leq t - 1$ for every $i, j \in S$ and $S$ is a maximal such set since if $i \in S$ and $j \notin S$ then $d(i, j) \geq t$. Thus $S$ is an equivalence class under the relation $i \sim j$ iff $d(i, j) \leq t - 1$ and so for every $i, j \in S$ we have $d(i, j) \leq |S| - 1$ or equivalently $x_{ij}^{|S|} = 0$. The ultrametric $d$ represented by a feasible solution $x_{ij}^t$ is given by $d(i, j) = \sum_{t=1}^{n-1} x_{ij}^t$.

**Definition 6.** *For any $\left\{x_{ij}^t \mid t \in [n-1], i, j \in V\right\}$ let $E_t$ be defined as $E_t := \left\{\{i,j\} \mid x_{ij}^t = 0\right\}$. Note that if $x_{ij}^t$ is feasible for ILP-ultrametric then $E_t \subseteq E_{t+1}$ for any $t$ since $x_{ij}^t \geq x_{ij}^{t+1}$. The sets $\{E_t\}_{t=1}^{n-1}$ induce a natural sequence of graphs $\{G_t\}_{t=1}^{n-1}$ where $G_t = (V, E_t)$ with $V$ being the data set.*

For a fixed $t \in \{1, \ldots, n-1\}$ it is instructive to study the combinatorial properties of the so called *layer-t problem*, where we fix a choice of $t$ and restrict ourselves to the constraints corresponding to that particular $t$. In particular we drop the inter-layer constraint (2), and constraints (3), (4) and (5) only range over $i, j, k \in V$ and $S \subseteq V$ with $t$ fixed. The following lemma provides a combinatorial characterization of feasible solutions to the layer-$t$ problem.

**Lemma 7.** *Fix a choice of $t \in [n-1]$. Let $G_t = (V, E_t)$ be the graph as in Definition 6 corresponding to a solution $x_{ij}^t$ to the layer-t problem. Then $G_t$ is a disjoint union of cliques of size $\leq t$. Moreover this exactly characterizes all feasible solutions to the layer-t ILP.*

By Lemma 7 the layer-$t$ problem is to find a subset $\overline{E}_t \subseteq E(K_n)$ of minimum weight under $\kappa$, such that the complement graph $G_t = (V, E_t)$ is a disjoint union of cliques of size $\leq t$. Our algorithmic approach is to solve an LP relaxation of ILP-ultrametric and then round the solution to get a feasible solution to ILP-ultrametric. The rounding however proceeds iteratively in a layer-wise manner and so we need to make sure that the rounded solution satisfies the inter-layer constraints (2) and (5). The following lemma gives a combinatorial characterization of solutions that satisfy these two constraints.

**Lemma 8.** *For every $t \in [n-1]$, let $x_{ij}^t$ be feasible for the layer-$t$ problem. Let $G_t = (V, E_t)$ be the graph as in Definition 6 corresponding to $x_{ij}^t$, so that by Lemma 7, $G_t$ is a disjoint union of cliques $K_1^t, \ldots, K_{l_t}^t$ each of size at most $t$. Then $x_{ij}^t$ is feasible for ILP-ultrametric if and only if the following conditions hold.*

**Nested cliques** *For any $s \leq t$ every clique $K_p^s$ for some $p \in [l_s]$ in $G_s$ is a subclique of some clique $K_q^t$ in $G_t$ where $q \in [l_t]$.*

**Realization** *If $\left| K_p^t \right| = s$ for some $s \leq t$, then $G_s$ contains $K_p^t$ as a component clique, i.e., $K_q^s = K_p^t$ for some $q \in [l_s]$.*

The combinatorial interpretation of the individual layer-$t$ problems allow us to simplify the formulation of ILP-ultrametric by replacing the constraints for sets of a specific size (Constraint 4) by a global constraint about all sets.

**Lemma 9.** *We may replace Constraint 4 of ILP-ultrametric by the following equivalent constraint $\sum_{j \in S} x_{ij}^t \geq |S| - t$, for every $t \in [n-1]$, $S \subseteq V$ and $i \in S$.*

# 4 Rounding an LP relaxation

In this section we consider the following natural LP relaxation for ILP-ultrametric. We keep the variables $x_{ij}^t$ for every $t \in [n-1]$ and $i, j \in V$ but relax the integrality constraint on the variables.

$$\min \quad \sum_{t=1}^{n-1} \sum_{\{i,j\} \in E(K_n)} \kappa(i,j) x_{ij}^t \qquad \text{(LP-ultrametric)}$$

$$\text{s.t.} \quad x_{ij}^t \geq x_{ij}^{t+1} \qquad \forall i, j \in V, t \in [n-2] \tag{8}$$

$$x_{ij}^t + x_{jk}^t \geq x_{ik}^t \qquad \forall i, j, k \in V, t \in [n-1] \tag{9}$$

$$\sum_{j \in S} x_{ij}^t \geq |S| - t \qquad \forall t \in [n-1], S \subseteq V, i \in S \tag{10}$$

$$x_{ij}^t = x_{ji}^t, x_{ii}^t = 0 \qquad \forall i, j \in V, t \in [n-1] \tag{11}$$

$$0 \leq x_{ij}^t \leq 1 \qquad \forall i, j \in V, t \in [n-1] \tag{12}$$

Note that the LP relaxation LP-ultrametric differs from ILP-ultrametric in not having constraint 5. A feasible solution $x_{ij}^t$ to LP-ultrametric induces a sequence $\{d_t\}_{t \in [n-1]}$ of distance metrics over $V$ defined as $d_t(i,j) := x_{ij}^t$. Constraint 10 enforces an additional restriction on this metric: informally points in a "large enough" subset $S$ should be spread apart according to the metric $d_t$. Metrics of type $d_t$ are called *spreading metrics* and were first studied by [19, 20] in relation to graph partitioning problems. The following lemma gives a technical interpretation of spreading metrics (see, e.g., [19, 20]).

**Lemma 10.** *Let $x_{ij}^t$ be feasible for LP-ultrametric and for a fixed $t \in [n-1]$, let $d_t$ be the induced spreading metric. Let $i \in V$ be an arbitrary vertex and let $S \subseteq V$ be a set containing $i$ such that $|S| > (1 + \varepsilon)t$ for some $\varepsilon > 0$. Then $\max_{j \in S} d_t(i,j) > \frac{\varepsilon}{1+\varepsilon}$.*

The following lemma states that we can optimize over LP-ultrametric in polynomial time.

**Lemma 11.** *An optimal solution to LP-ultrametric can be computed in time polynomial in $n$ and $\log\left(\max_{i,j} \kappa(i,j)\right)$.*

From now on we will simply refer to a feasible solution of LP-ultrametric by the sequence of spreading metrics $\{d_t\}_{t \in [n-1]}$ it induces. The following definition introduces the notion of an open

ball $\mathcal{B}_U(i, r, t)$ of radius $r$ centered at $i \in V$ according to the metric $d_t$ and restricted to the set $U \subseteq V$.

**Definition 12.** *Let $\{d_t \mid t \in [n-1]\}$ be the sequence of spreading metrics feasible for [LP-ultrametric](#). Let $U \subseteq V$ be an arbitrary subset of $V$. For a vertex $i \in U$, $r \in \mathbb{R}$, and $t \in [n-1]$ we define the* open ball $\mathcal{B}_U(i, r, t)$ *of radius $r$ centered at $i$ as*

$$\mathcal{B}_U(i, r, t) := \{j \in U \mid d_t(i, j) < r\} \subseteq U.$$

*If $U = V$ then we denote $\mathcal{B}_U(i, r, t)$ simply by $\mathcal{B}(i, r, t)$.*

To round [LP-ultrametric](#) to get a feasible solution for [ILP-ultrametric](#), we will use the technique of *sphere growing* which was introduced in [33] to show an $O(\log n)$ approximation for the maximum multicommodity flow problem. The basic idea is to grow a ball around a vertex until the expansion of this ball is below a certain threshold, chop off this ball and declare it as a partition and recurse on the remaining vertices. Since then this idea has been used by [25, 19, 14] to design approximation algorithms for various graph partitioning problems. The first step is to associate to every ball $\mathcal{B}_U(i, r, t)$ a volume $\mathrm{vol}(\mathcal{B}_U(i, r, t))$ and a boundary $\partial\mathcal{B}_U(i, r, t)$ so that its expansion is defined. For any $t \in [n-1]$ and $U \subseteq V$ we denote by $\gamma_t^U$ the value of the layer-$t$ objective for solution $d_t$ restricted to the set $U$, i.e., $\gamma_t^U := \sum_{\substack{i, j \in U \\ i < j}} \kappa(i, j) d_t(i, j)$. When $U = V$ we refer to $\gamma_t^U$ simply by $\gamma_t$. Since $\kappa : V \times V \to \mathbb{R}_{\geq 0}$, it follows that $\gamma_t^U \leq \gamma_t$ for any $U \subseteq V$. We are now ready to define the volume, boundary and expansion of a ball $\mathcal{B}_U(i, r, t)$. We use the definition of [19] modified for restrictions to arbitrary subsets $U \subseteq V$.

**Definition 13.** *[19] Let $U$ be an arbitrary subset of $V$. For a vertex $i \in U$, radius $r \in \mathbb{R}$, and $t \in [n-1]$, let $\mathcal{B}_U(i, r, t)$ be the ball of radius $r$ as in Definition 12. Then we define its* volume *as*

$$\mathrm{vol}(\mathcal{B}_U(i, r, t)) := \frac{\gamma_t^U}{n \log n} + \sum_{\substack{j, k \in \mathcal{B}_U(i, r, t) \\ j < k}} \kappa(j, k) d_t(j, k) + \sum_{\substack{j \in \mathcal{B}_U(i, r, t) \\ k \notin \mathcal{B}_U(i, r, t) \\ k \in U}} \kappa(j, k)(r - d_t(i, j)).$$

*The* boundary *of the ball $\partial\mathcal{B}_U(i, r, t)$ is the partial derivative of volume with respect to the radius, i.e., $\partial\mathcal{B}_U(i, r, t) := \frac{\partial \mathrm{vol}(\mathcal{B}_U(i, r, t))}{\partial r}$. The* expansion $\phi(\mathcal{B}_U(i, r, t))$ *of the ball $\mathcal{B}_U(i, r, t)$ is then defined as the ratio of its boundary to its volume, i.e., $\phi(\mathcal{B}_U(i, r, t)) := \frac{\partial\mathcal{B}_U(i, r, t)}{\mathrm{vol}(\mathcal{B}_U(i, r, t))}$.*

The following theorem establishes that the rounding procedure of Algorithm 1 ensures that the cliques in $\mathcal{C}_t$ are "small" and that the cost of the edges removed to form them are not too high. It also shows that Algorithm 1 can be implemented to run in time polynomial in $n$. Let $m_\varepsilon := \left\lfloor \frac{n-1}{1+\varepsilon} \right\rfloor$ as in Algorithm 1.

**Theorem 14.** *Let $\{x_{ij}^t \mid t \in [m_\varepsilon], i, j \in V\}$ be the output of Algorithm 1 on a feasible solution $\{d_t\}_{t \in [n-1]}$ of [LP-ultrametric](#) and any choice of $\varepsilon \in (0, 1)$. For any $t \in [m_\varepsilon]$, $x_{ij}^t$ is feasible for the layer-$\lfloor (1+\varepsilon)t \rfloor$ problem and there is a constant $c(\varepsilon) > 0$ depending only on $\varepsilon$ such that $\sum_{\{i,j\} \in E(K_n)} \kappa(i, j) x_{ij}^t \leq c(\varepsilon)(\log n)\gamma_t$. Moreover, Algorithm 1 can be implemented to run in time polynomial in $n$.*

We are now ready to state the main theorem showing that we can obtain a low cost non-trivial ultrametric from Algorithm 1. The proof idea of the main theorem is to use the combinatorial characterization of Lemma 8 to show that the rounded solution is feasible for [ILP-ultrametric](#) besides using Theorem 14 for the individual layer-$t$ guarantees.

**Theorem 15.** *Let $\{x_{ij}^t \mid t \in [m_\varepsilon], i, j \in V\}$ be the output of Algorithm 1 on an optimal solution $\{d_t\}_{t \in [n-1]}$ of [LP-ultrametric](#) for any choice of $\varepsilon \in (0, 1)$. Define the sequence $\{y_{ij}^t\}$ for every $t \in [n-1]$ and $i, j \in V$ as $y_{ij}^t := x_{ij}^{\lfloor t/(1+\varepsilon) \rfloor}$ if $t > 1 + \varepsilon$ and $y_{ij}^t := 1$ otherwise. Then $y_{ij}^t$ is feasible for [ILP-ultrametric](#) and satisfies $\sum_{t=1}^{n-1} \sum_{\{i,j\} \in E(K_n)} \kappa(i, j) y_{ij}^t \leq (2c(\varepsilon) \log n) \mathrm{OPT}$, where $\mathrm{OPT}$ is the optimal solution to [ILP-ultrametric](#) and $c(\varepsilon)$ is the constant in the statement of Theorem 14.*

Lemma 11 and Theorem 15 imply the following corollary where we put everything together to obtain a hierarchical clustering of $V$ in time polynomial in $n$ with $|V| = n$. Let $\mathcal{T}$ denote the set of all possible hierarchical clusterings of $V$.

**Algorithm 1:** Iterative rounding algorithm to find a low cost ultrametric

---

**Input**: Data set $V$, $\{d_t\}_{t\in[n-1]} : V \times V$, $\varepsilon > 0$, $\kappa : V \times V \to \mathbb{R}_{\geq 0}$

**Output**: A solution set of the form $\left\{ x_{ij}^t \in \{0,1\} \mid t \in \left[\left\lfloor \frac{n-1}{1+\varepsilon} \right\rfloor\right], i,j \in V \right\}$

$m_\varepsilon \leftarrow \left\lfloor \frac{n-1}{1+\varepsilon} \right\rfloor$
$t \leftarrow m_\varepsilon$
$\mathcal{C}_{t+1} \leftarrow \{V\}$
$\Delta \leftarrow \frac{\varepsilon}{1+\varepsilon}$
**while** $t \geq 1$ **do**
    $\mathcal{C}_t \leftarrow \emptyset$
    **for** $U \in \mathcal{C}_{t+1}$ **do**
        **if** $|U| \leq (1+\varepsilon)t$ **then**
            $\mathcal{C}_t \leftarrow \mathcal{C}_t \cup \{U\}$
            Go to line 1
        **end**
        **while** $U \neq \emptyset$ **do**
            Let $i$ be arbitrary in $U$
            Let $r \in (0,\Delta]$ be s.t. $\phi\left(\mathcal{B}_U\left(i,r,t\right)\right) \leq \frac{1}{\Delta} \log\left(\frac{\text{vol}(\mathcal{B}_U(i,\Delta,t))}{\text{vol}(\mathcal{B}_U(i,0,t))}\right)$
            $\mathcal{C}_t \leftarrow \mathcal{C}_t \cup \{\mathcal{B}_U\left(i,r,t\right)\}$
            $U \leftarrow U \setminus \mathcal{B}_U\left(i,r,t\right)$
        **end**
    **end**
    $x_{ij}^t = 1$ if $i \in U_1 \in \mathcal{C}_t$, $j \in U_2 \in C_t$ and $U_1 \neq U_2$, else $x_{ij}^t = 0$
    $t \leftarrow t - 1$
**end**
**return** $\left\{ x_{ij}^t \mid t \in [m_\varepsilon], i,j \in V \right\}$

---

**Corollary 16.** *Given a data set $V$ of $n$ points and a similarity function $\kappa : V \times V \to \mathbb{R}_{\geq 0}$, there is an algorithm to compute a hierarchical clustering $T$ of $V$ satisfying $\text{cost}(T) \leq O(\log n) \min_{T' \in \mathcal{T}} \text{cost}(T')$ in time polynomial in $n$ and $\log(\max_{i,j \in V} \kappa(i,j))$.*

## 5   Generalized Cost Function

In this section we study the following natural generalization of cost function (1) also introduced by [16] where the distance between the two points is scaled by a function $f : \mathbb{R}_{\geq 0} \to \mathbb{R}_{\geq 0}$ i.e., $\text{cost}_f(T) \coloneqq \sum_{\{i,j\} \in E(K_n)} \kappa(i,j) f(|\text{leaves } T[\text{lca}(i,j)]|)$. In order for this cost function to make sense, $f$ should be strictly increasing and satisfy $f(0) = 0$. Possible choices for $f$ could be in $\{x^2, e^x - 1, \log(1+x)\}$. The top-down heuristic in [16] finds the optimal hierarchical clustering up to an approximation factor of $c_n \log n$ with $c_n$ being defined as $c_n \coloneqq 3\alpha_n \max_{1 \leq n' \leq n} \frac{f(n')}{f(\lceil n'/3\rceil)}$ and where $\alpha_n$ is the approximation factor from the Sparsest Cut algorithm used.

A naive approach to solving this problem using the ideas of Algorithm 1 would be to replace the objective function of ILP-ultrametric by $\sum_{\{i,j\} \in E(K_n)} \kappa(i,j) f\left(\sum_{t=1}^{n-1} x_{ij}^t\right)$. This makes the corresponding analogue of LP-ultrametric non-linear however, and for a general $\kappa$ and $f$ it is not clear how to compute an optimum solution in polynomial time. Using a small trick, one can still prove that Algorithm 1 returns a good approximation in this case as the following theorem states. For more details on the generalized cost function we refer the reader to the supplementary material.

**Theorem 17.** *Let $a_n \coloneqq \max_{n' \in [n]}(f(n') - f(n'-1))$. Given a data set $V$ of $n$ points and a similarity function $\kappa : V \times V \to \mathbb{R}_{\geq 0}$, there is an algorithm to compute a hierarchical clustering $T$ of $V$ satisfying $\text{cost}_f(T) \leq \bar{O}(\log n + a_n) \min_{T' \in \mathcal{T}} \text{cost}_f(T')$ in time polynomial in $n$, $\log(\max_{i,j \in V} \kappa(i,j))$ and $\log f(n)$.*

Note that, in this case we pay a price of $O(\log f(n))$ in the running time due to binary search.

## 6 Experiments

Finally, we describe the experiments we performed. We implemented a generalized version of ILP-ultrametric where one can plug in any strictly increasing function $f$ satisfying $f(0) = 0$. For the sake of exposition, we limited ourselves to $\{x, x^2, \log(1+x), e^x - 1\}$ for the function $f$. We used the *dual simplex* method and separate constraints (9) and (10) to obtain fast computations. For the similarity function $\kappa$ we limited ourselves to using *cosine similarity* $\kappa_{cos}$ and the *Gaussian kernel* $\kappa_{gauss}$ with $\sigma = 1$. Since Algorithm 1 requires $\kappa \geq 0$, in practice we use $1 + \kappa_{cos}$ instead of $\kappa_{cos}$. Note that both Ward's method and the $k$-means algorithm work on the squared Euclidean distance and thus need vector representations of the data set. For the linkage based algorithms we use the same similarity function that we use for Algorithm 1.

We considered synthetic data sets and some data sets from the UCI database [36]. The synthetic data sets were mixtures of Gaussians in various small dimensional spaces and for some of the larger data sets we subsampled a smaller number of points uniformly at random for a number of times depending on the performance of the MIP and LP solver. For a comparison of the cost of the hierarchy returned by Algorithm 1 and the optimal hierarchy obtained by solving ILP-ultrametric, see the supplementary material.

To compare the different hierarchical clustering algorithms, we prune the hierarchy to get the *best $k$* flat clusters and measure its error relative to the ground truth. We use the following notion of error also known as *Classification Error* that is standard in the literature for hierarchical clustering (see, e.g., [37]).

**Definition 18.** *Given a proposed clustering $h : V \to \{1, \ldots, k\}$ its* classification error *relative to a target clustering $g : V \to \{1, \ldots, k\}$ is denoted by* $\mathrm{err}\,(g, h)$ *and is defined as* $\mathrm{err}\,(g, h) := \min_{\sigma \in S_k} \left[ \Pr_{x \in V}[h(x) \neq \sigma(g(x))] \right].$

Figure 1 shows that Algorithm 1 often gives better prunings compared to the other standard clustering algorithms with respect to this notion of error.

## 7 Conclusion

In this work we have studied the cost function introduced by [16] for hierarchical clustering of data under a pairwise similarity function. We have shown a combinatorial characterization of ultrametrics induced by this cost function leading to an improved approximation algorithm for this problem. It remains for future work to investigate combinatorial algorithms for this cost function as well as algorithms for other cost functions of a similar flavor; see supplementary material for a discussion.

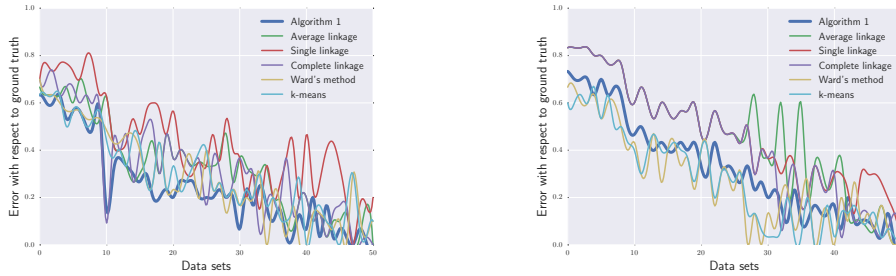

Figure 1: Comparison of Algorithm 1 with other algorithms for clustering using $1 + \kappa_{cos}$ (left) and $\kappa_{gauss}$ (right)

## 8 Acknowledgments

Research reported in this paper was partially supported by NSF CAREER award CMMI-1452463 and NSF grant CMMI-1333789. The authors thank Kunal Talwar and Mohit Singh for helpful discussions and anonymous reviewers for helping improve the presentation of this paper.

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
