[Supplementary Material · supplemental.pdf]

# Hierarchical Clustering via Spreading Metrics

Aurko Roy[1] and Sebastian Pokutta[2]

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

Besides this prior work on hierarchical clustering we are also motivated by the long line of work in the classical clustering setting where a popular strategy is to study convex relaxations of these problems and to round an optimal fractional solution into an integral one with the aim of getting a good approximation to the cost function. A long line of work (see, e.g., [13, 29, 28, 15]) has employed this approach on LP relaxations for the $k$-median problem, including [35] which gives the best known approximation factor of $1 + \sqrt{3} + \varepsilon$. Similarly, a few authors have studied LP and SDP relaxations for the $k$-means problem (see, e.g., [39, 38, 4]), while one of the best known algorithms for kernel $k$-means and spectral clustering is due to [42] which approximates the nonnegative matrix factorization (NMF) problem by LPs.

LP relaxations for hierarchical clustering have also been studied in [2] where the objective is to fit a tree metric to a data set given pairwise dissimilarities. While the LP relaxation and rounding algorithm in [2] is similar in flavor, the result is incomparable to ours (see Section 7 for a discussion). Another work that is indirectly related to our approach is [18] where the authors study an ILP to obtain a closest ultrametric to arbitrary functions on a discrete set. Our approach is to give a combinatorial characterization of the ultrametrics induced by the cost function of [16] which allows us to use the tools from [18] to model the problem as an ILP. The natural LP relaxation of this ILP turns out to be closely related to LP relaxations considered before for several graph partitioning problems (see, e.g., [33, 34, 19, 32]) and we use a rounding technique studied in this context to round this LP relaxation.

Recently, we became aware of independent work by [12] obtaining similar results for hierarchical clustering. In particular [12] improve the approximation factor to $O\left(\sqrt{\log n}\right)$ by showing how to round a spreading metric SDP relaxation for this cost function. The analysis of this rounding procedure also enabled them to show that the top-down heuristic of [16] actually returns an $O(\sqrt{\log n})$ approximate clustering rather than an $O\left(\log^{3/2} n\right)$ approximate clustering. They also analyzed a very similar LP relaxation using the *divide-and-conquer approximation algorithms using spreading metrics* paradigm of [20] together with a result of [7] to show an $O(\log n)$ approximation. Finally, they also gave similar constant factor inapproximability results for this problem.

## 1.2 Contribution

While studying convex relaxations of optimization problems is fairly natural, for the cost function introduced in [16] however, it is not immediately clear how one would go about writing such a relaxation. Our first contribution is to give a combinatorial characterization of the family of ultrametrics induced by this cost function on hierarchies. Inspired by the approach in [18] where the authors study an integer linear program for finding the closest ultrametric, we are able to formulate the problem of finding the minimum cost hierarchical clustering as an integer linear program. Interestingly and perhaps unsurprisingly, the specific family of ultrametrics induced by this cost function give rise to linear constraints studied before in the context of finding balanced separators in weighted graphs. We then show how to round an optimal fractional solution using the *sphere growing* technique first introduced in [33] (see also [25, 19, 14]) to recover a tree of cost at most $O(\log n)$ times the optimal tree for this cost function. The generalization of this cost function involves scaling every pairwise distances by an arbitrary strictly increasing function $f$ satisfying $f(0) = 0$. We modify the integer linear program for this general case and show that the rounding algorithm still finds a hierarchical clustering of cost at most $O(\log n)$ times the optimal clustering in this setting. We also show a constant factor inapproximability result for this problem for any polynomial sized LP and SDP relaxations and under the assumption of the *Small Set Expansion* hypothesis. We conclude with an experimental study of the integer linear program and the rounding algorithm on some synthetic and real world data sets to show that the approximation algorithm often recovers clusters close to the true optimum (according to this cost function) and that its projections into flat clusters often has a better error rate than the linkage based algorithms and the $k$-means algorithm.

## 2 Preliminaries

A similarity based clustering problem consists of a dataset $V$ of $n$ points and a *similarity function* $\kappa : V \times V \to \mathbb{R}_{\geq 0}$ such that $\kappa(i, j)$ is a measure of the similarity between $i$ and $j$ for any $i, j \in V$. We will assume that the similarity function is symmetric i.e., $\kappa(i, j) = \kappa(j, i)$ for every $i, j \in V$. Note that we do not make any assumptions about the points in $V$ coming from an underlying metric space. For a given instance of a clustering problem we have an associated weighted complete graph $K_n$ with vertex set $V$ and weight function given by $\kappa$. A *hierarchical clustering* of $V$ is a tree $T$ with a designated root $r$ and with the elements of $V$ as its leaves, i.e., $\text{leaves}(T) = V$. For any set $S \subseteq V$ we denote the *lowest common ancestor* of $S$ in $T$ by $\text{lca}(S)$. For pairs of points $i, j \in V$ we will abuse the notation for the sake of simplicity and denote $\text{lca}(\{i, j\})$ simply by $\text{lca}(i, j)$. For a node $v$ of $T$ we denote the subtree of $T$ rooted at $v$ by $T[v]$. The following cost function was introduced by [16] to measure the quality of the hierarchical clustering $T$

$$\text{cost}(T) := \sum_{\{i,j\} \in E(K_n)} \kappa(i, j) \left|\text{leaves}(T[\text{lca}(i, j)])\right|. \tag{1}$$

The intuition behind this cost function is as follows. Let $T$ be a hierarchical clustering with designated root $r$ so that $r$ represents the whole data set $V$. Since $\text{leaves}(T) = V$, every internal node $v \in T$ represents a cluster of its descendant leaves, with the leaves themselves representing singleton clusters of $V$. Starting from $r$ and going down the tree, every distinct pair of points $i, j \in V$ will be eventually separated at the leaves. If $\kappa(i, j)$ is large, i.e., $i$ and $j$ are very similar to each other then we would like them to be separated as far down the tree as possible if $T$ is a good clustering of $V$. This is enforced in the cost function (1): if $\kappa(i, j)$ is large then the number of leaves of $\text{lca}(i, j)$ should be small i.e., $\text{lca}(i, j)$ should be far from the root $r$ of $T$. Such a cost function is not unique however; see Section 7 for some other cost functions of a similar flavor.

Note that while requiring $\kappa$ to be non-negative might seem like an artificial restriction, cost function (1) breaks down when all the $\kappa(i,j) < 0$, since in this case the trivial clustering $r, T^*$ where $T^*$ is the star graph with $V$ as its leaves is always the minimizer. Therefore in the rest of this work we will assume that $\kappa \geq 0$. This is not a restriction compared to [16], since the Sparsest Cut algorithm used as a subroutine also requires this assumption. Let us now briefly recall the notion of an ultrametric.

**Definition 1** (Ultrametric). *An* ultrametric *on a set $X$ of points is a distance function $d : X \times X \to \mathbb{R}$ satisfying the following properties for every $x, y, z \in X$*

1. *Nonnegativity: $d(x,y) \geq 0$ with $d(x,y) = 0$ iff $x = y$*

2. *Symmetry: $d(x,y) = d(y,x)$*

3. *Strong triangle inequality: $d(x,y) \leq \max\{d(y,z), d(z,x)\}$*

Under the cost function (1), one can interpret the tree $T$ as inducing an ultrametric $d_T$ on $V$ given by $d_T(i,j) := |\text{leaves}(T[\text{lca}\,(i,j)])| - 1$. This is an ultrametric since $d_T(i,j) = 0$ iff $i = j$ and for any triple $i, j, k \in V$ we have $d_T(i,j) \leq \max\{d_T(i,k), d_T(j,k)\}$. The following definition introduces the notion of *non-trivial ultrametrics*. These turn out to be precisely the ultrametrics that are induced by tree decompositions of $V$ corresponding to cost function (1), as we will show in Corollary 8.

**Definition 2.** *An ultrametric $d$ on a set of points $V$ is* non-trivial *if the following conditions hold.*

1. *For every non-empty set $S \subseteq V$, there is a pair of points $i, j \in S$ such that $d(i,j) \geq |S| - 1$.*

2. *For any $t$ if $S_t$ is an equivalence class of $V$ under the relation $i \sim j$ iff $d(i,j) \leq t$, then $\max_{i,j \in S_t} d(i,j) \leq |S_t| - 1$.*

Note that for an equivalence class $S_t$ where $d(i,j) \leq t$ for every $i, j \in S_t$ it follows from Condition 1 that $t \geq |S_t| - 1$. Thus in the case when $t = |S_t| - 1$ the two conditions imply that the maximum distance between any two points in $S$ is $t$ and that there is a pair $i, j \in S$ for which this maximum is attained. The following lemma shows that non-trivial ultrametrics behave well under restrictions to equivalence classes $S_t$ of the form $i \sim j$ iff $d(i,j) \leq t$.

**Lemma 3.** *Let $d$ be a non-trivial ultrametric on $V$ and let $S_t \subseteq V$ be an equivalence class under the relation $i \sim j$ iff $d(i,j) \leq t$. Then $d$ restricted to $S_t$ is a non-trivial ultrametric on $S_t$.*

*Proof.* Clearly $d$ restricted to $S_t$ is an ultrametric on $S_t$ and so we need to establish that it satisfies Conditions 1 and 2 of Definition 2. Let $S \subseteq S_t$ be any set. Since $d$ is a non-trivial ultrametric on $V$ it follows that there is a pair $i, j \in S$ with $d(i,j) \geq |S| - 1$, and so $d$ restricted to $S_t$ satisfies Condition 1.

If $S'_r$ is an equivalence class in $S_t$ under the relation $i \sim j$ iff $d(i,j) \leq r$ then clearly $S'_r = S_t$ if $r > t$. Since $d$ is a non-trivial ultrametric on $V$, it follows that $\max_{i,j \in S'_r} d(i,j) = \max_{i,j \in S_t} d(i,j) \leq |S_t| - 1 = |S'_r| - 1$. Thus we may assume that $r \leq t$. Consider an $i \in S'_r$ and let $j \in V$ be such that $d(i,j) \leq r$. Since $r \leq t$ and $i \in S_t$, it follows that $j \in S_t$ and so $j \in S'_r$. In other words $S'_r$ is an equivalence class in $V$ under the relation $i \sim j$ iff $d(i,j) \leq r$. Since $d$ is an ultrametric on $V$ it follows that $\max_{i,j \in S'_r} d(i,j) \leq |S'_r| - 1$. Thus $d$ restricted to $S_t$ satisfies Condition 2. $\qquad\square$

The intuition behind the two conditions in Definition 2 is as follows. Condition 1 imposes a certain lower bound by ruling out trivial ultrametrics where, e.g., $d(i,j) = 1$ for every distinct pair $i, j \in V$. On the other hand Condition 2 discretizes and imposes an upper bound on $d$ by restricting its range to the set $\{0, 1, \ldots, n - 1\}$ (see Lemma 4). This rules out the other spectrum of triviality where for example $d(i,j) = n$ for every distinct pair $i, j \in V$ with $|V| = n$.

**Lemma 4.** *Let $d$ be a non-trivial ultrametric on the set $V$ as in Definition 2. Then the range of $d$ is contained in the set $\{0, 1, \ldots, n - 1\}$ with $|V| = n$.*

*Proof.* We will prove this by induction on $|V|$. The base case when $|V| = 1$ is trivial. Therefore, we now assume that $|V| > 1$. By Condition 1 there is a pair $i, j \in V$ such that $d(i,j) \geq n - 1$. Let $t = \max_{i,j \in V} d(i,j)$, then the only equivalence class under the relation $i \sim j$ iff $d(i,j) \leq t$ is $V$. By Condition 2 it follows that $\max_{i,j \in V} d(i,j) = t = n - 1$. Let $V_1, \ldots V_m$ denote the

set of equivalence classes of $V$ under the relation $i \sim j$ iff $d(i, j) \leq n - 2$. Note that $m > 1$ as there is a pair $i, j \in V$ with $d(i, j) = n - 1$, and therefore each $V_l \subsetneq V$. By Lemma 3, $d$ restricted to each of these $V_i$'s is a non-trivial ultrametric on those sets. The claim then follows immediately: for any $i, j \in V$ either $i, j \in V_l$ for some $V_l$ in which case by the induction hypothesis $d(i, j) \in \{0, 1, \dots, |V_l| - 1\}$, or $i \in V_l$ and $j \in V_{l'}$ for $l \neq l'$ in which case $d(i, j) = n - 1$. $\qquad \square$

## 3 Ultrametrics and Hierarchical Clusterings

We start with the following easy lemma about the lowest common ancestors of subsets of $V$ in a hierarchical clustering $T$ of $V$.

**Lemma 5.** *Let $S \subseteq V$ with $|S| \geq 2$. If $r = \mathrm{lca}(S)$ then there is a pair $i, j \in S$ such that $\mathrm{lca}(i, j) = r$.*

*Proof.* We will proceed by induction on $|S|$. If $|S| = 2$ then the claim is trivial and so we may assume $|S| > 2$. Let $i \in S$ be an arbitrary point and let $r' = \mathrm{lca}(S \setminus \{i\})$. We claim that $r = \mathrm{lca}(i, r')$. Clearly the subtree rooted at $\mathrm{lca}(i, r')$ contains $S$ and since $T[r]$ is the smallest such tree it follows that $r \in T[\mathrm{lca}(i, r')]$.

Conversely, $T[r]$ contains $S \setminus \{i\}$ and so $r' \in T[r]$ and since $i \in T[r]$, it follows that $\mathrm{lca}(i, r') \in T[r]$. Thus we conclude that $r = \mathrm{lca}(i, r')$.

If $\mathrm{lca}(i, r') = r'$, then we are done by the induction hypothesis. Thus we may assume that $i \notin T[r']$. Consider any $j \in S$ such that $j \in T[r']$. Then we have that $\mathrm{lca}(i, j) = r$ as $\mathrm{lca}(i, r') = r$ and $j \in T[r']$ and $i \notin T[r']$. $\qquad \square$

We will now show that non-trivial ultrametrics on $V$ as in Definition 2 are exactly those that are induced by hierarchical clusterings on $V$ under cost function (1). The following lemma shows the forward direction: the ultrametric $d_T$ induced by any hierarchical clustering $T$ is non-trivial.

**Lemma 6.** *Let $T$ be a hierarchical clustering on $V$ and let $d_T$ be the ultrametric on $V$ induced by it. Then $d_T$ is non-trivial.*

*Proof.* Let $S \subseteq V$ be arbitrary and $r = \mathrm{lca}(S)$, then $T[r]$ has at least $|S|$ leaves. By Lemma 5 there must be a pair $i, j \in S$ such that $r = \mathrm{lca}(i, j)$ and so $d_T(i, j) \geq |S| - 1$. This satisfies Condition 1 of non-triviality.

For any $t$, let $S_t$ be a non-empty equivalence class under the relation $i \sim j$ iff $d_T(i, j) \leq t$. Since $d_T$ satisfies Condition 1 it follows that $|S_t| - 1 \leq t$. Let us assume for the sake of contradiction that there is a pair $i, j \in S_t$ such that $d_T(i, j) > |S_t| - 1$. Let $r = \mathrm{lca}(S_t)$; using the definition of $d_T$ it follows that $t + 1 \geq |\mathrm{leaves}(T[r])| > |S_t|$ since $i, j \in S_t$. Let $k \in \mathrm{leaves}(T[r]) \setminus S_t$ be an arbitrary point, then for every $l \in S_t$ it follows that $d_T(k, l) \leq |\mathrm{leaves}(T[r])| - 1 \leq t$ since the subtree rooted at $r$ contains both $k$ and $l$. This is a contradiction to $S_t$ being an equivalence class under $i \sim j$ iff $d_T(i, j) \leq t$ since $k \notin S_t$. Thus $d_T$ also satisfies Condition 2 of Definition 2. $\qquad \square$

The following crucial lemma shows the converse: every non-trivial ultrametric on $V$ is realized by a hierarchical clustering $T$ of $V$.

**Lemma 7.** *For every non-trivial ultrametric $d$ on $V$ there is a hierarchical clustering $T$ on $V$ such that for any pair $i, j \in V$ we have*

$$d_T(i, j) = |\mathrm{leaves}(T[\mathrm{lca}(i, j)])| - 1 = d(i, j).$$

*Moreover this hierarchy can be constructed in time $O(n^3)$ by Algorithm 1 where $|V| = n$.*

*Proof.* The proof is by induction on $n$. The base case when $n = 1$ is straightforward. We now suppose that the statement is true for sets of size $< n$. Note that $i \sim j$ iff $d(i, j) \leq n - 2$ is an equivalence relation on $V$ and thus partitions $V$ into $m$ equivalence classes $V_1, \dots, V_m$. We first observe that $m > 1$ since by Condition 1 there is a pair of points $i, j \in V$ such that $d(i, j) \geq n - 1$ and in particular $|V|_l < n$ for every $l \in \{1, \dots, m\}$. By Lemma 3, $d$ restricted to any $V_l$ is a non-trivial ultrametric on $V_l$ and there is a pair of points $i, j \in V_l$ such that $d(i, j) = |V_l| - 1$ by Conditions 1 and 2. Therefore by the induction hypothesis we construct trees $T_1, \dots, T_m$ such that

for every $l \in \{1, \ldots, m\}$ we have $\mathrm{leaves}(T_l) = V_l$. Further for any pair of points $i, j \in V_l$ for some $l \in \{1, \ldots, m\}$, we also have $d(i, j) = d_{T_l}(i, j)$.

We construct the tree $T$ as follows: we first add a root $r$ and then connect the root $r_l$ of $T_l$ to $r$ for every $l \in \{1, \ldots, m\}$. Consider a pair of points $i, j \in V$. If $i, j \in V_l$ for some $l \in \{1, \ldots, m\}$ then we are done since $d_{T_l}(i, j) = d_T(i, j)$ as $\mathrm{lca}(i, j) \in T_l$. If $i \in V_l$ and $j \in V_{l'}$ for some $l \neq l'$ then $d(i, j) = n - 1$ since $d(i, j) \geq n - 1$ by definition of the equivalence relation and the range of $d$ lies in $\{0, 1, \ldots, n-1\}$ by Lemma 4. Moreover $i$ and $j$ are leaves in $T_l$ and $T_{l'}$ respectively, and thus by construction of $T$ we have $\mathrm{lca}(i, j) = r$, i.e., $d_T(i, j) = n - 1$ and so the claim follows. Algorithm 1 simulates this inductive argument can be easily implemented to run in time $O\left(n^3\right)$. □

Lemmas 6 and 7 together imply the following corollary about the equivalence of hierarchical clusterings and non-trivial ultrametrics.

**Corollary 8.** *There is a bijection between the set of hierarchical clusterings $T$ on $V$ and the set of non-trivial ultrametrics $d$ on $V$ satisfying the following conditions.*

1. *For every hierarchical clustering $T$ on $V$, there is a non-trivial ultrametric $d_T$ defined as $d_T(i, j) := |\mathrm{leaves}\, T[\mathrm{lca}(i, j)]| - 1$ for every $i, j \in V$.*

2. *For every non-trivial ultrametric $d$ on $V$, there is a hierarchical clustering $T$ on $V$ such that for every $i, j \in V$ we have $|\mathrm{leaves}\, T[\mathrm{lca}(i, j)]| - 1 = d(i, j)$.*

*Moreover this bijection can be computed in $O(n^3)$ time, where $|V| = n$.*

---

**Algorithm 1:** Hierarchical clustering of $V$ from non-trivial ultrametric

---

**Input**: Data set $V$ of $n$ points, non-trivial ultrametric $d : V \times V \to \mathbb{R}_{\geq 0}$
**Output**: Hierarchical clustering $T$ of $V$ with root $r$

**1** $r \leftarrow$ arbitrary choice of designated root in $V$
**2** $X \leftarrow \{r\}$
**3** $E \leftarrow \emptyset$
**4** **if** $n = 1$ **then**
**5** $\quad$ $T \leftarrow (X, E)$
**6** $\quad$ **return** $r, T$
**7** **else**
**8** $\quad$ Partition $V$ into $\{V_1, \ldots V_m\}$ under the equivalence relation $i \sim j$ iff $d(i, j) < n - 1$
**9** $\quad$ **for** $l \in \{1, \ldots, m\}$ **do**
**10** $\quad\quad$ Let $r_l, T_l$ be output of Algorithm 1 on $V_l, d|_{V_l}$
**11** $\quad\quad$ $X \leftarrow X \cup V(T_l)$
**12** $\quad\quad$ $E \leftarrow E \cup \{r, r_l\}$
**13** $\quad$ **end**
**14** $\quad$ $T \leftarrow (X, E)$
**15** $\quad$ **return** $r, T$
**16** **end**

---

Therefore to find the hierarchical clustering of minimum cost, it suffices to minimize $\langle \kappa, d \rangle$ over non-trivial ultrametrics $d : V \times V \to \{0, \ldots, n - 1\}$, where $V$ is the data set. Note that the cost of the ultrametric $d_T$ corresponding to a tree $T$ is an affine offset of $\mathrm{cost}(T)$. In particular, we have $\langle \kappa, d_T \rangle = \mathrm{cost}(T) - \sum_{\{i,j\} \in E(K_n)} \kappa(i, j)$.

A natural approach is to formulate this problem as an Integer Linear Program (ILP) and then study LP or SDP relaxations of it. We consider the following ILP for this problem that is motivated by [18]. We have the variables $x_{ij}^1, \ldots, x_{ij}^{n-1}$ for every distinct pair $i, j \in V$ with $x_{ij}^t = 1$ if and only if $d(i, j) \geq t$. For any positive integer $n$, let $[n] := \{1, 2, \ldots, n\}$.

$$\min \quad \sum_{t=1}^{n-1} \sum_{\{i,j\} \in E(K_n)} \kappa(i, j) x_{ij}^t \qquad \text{(ILP-ultrametric)}$$

$$\text{s.t.} \quad x_{ij}^t \geq x_{ij}^{t+1} \qquad \forall i,j \in V, t \in [n-2] \tag{2}$$

$$x_{ij}^t + x_{jk}^t \geq x_{ik}^t \qquad \forall i,j,k \in V, t \in [n-1] \tag{3}$$

$$\sum_{i,j \in S} x_{ij}^t \geq 2 \qquad \forall t \in [n-1], S \subseteq V, |S| = t+1 \tag{4}$$

$$\sum_{i,j \in S} x_{ij}^{|S|} \leq |S|^2 \left( \sum_{i,j \in S} x_{ij}^t + \sum_{\substack{i \in S \\ j \notin S}} \left(1 - x_{ij}^t\right) \right) \forall t \in [n-1], S \subseteq V \tag{5}$$

$$x_{ij}^t = x_{ji}^t \qquad \forall i,j \in V, t \in [n-1] \tag{6}$$

$$x_{ii}^t = 0 \qquad \forall i \in V, t \in [n-1] \tag{7}$$

$$x_{ij}^t \in \{0,1\} \qquad \forall i,j \in V, t \in [n-1] \tag{8}$$

Constraints (2) and (7) follow from the interpretation of the variables $x_{ij}^t$: if $d(i,j) \geq t$, i.e., $x_{ij}^t = 1$ then clearly $d(i,j) \geq t-1$ and so $x_{ij}^{t-1} = 1$. Furthermore, for any $i \in V$ we have $d(i,i) = 0$ and so $x_{ii}^t = 0$ for every $t \in [n-1]$. Note that constraint (3) is the same as the *strong triangle inequality* (Definition 1) since the variables $x_{ij}^t$ are in $\{0,1\}$. Constraint 6 ensures that the ultrametric is symmetric. Constraint 4 ensures the ultrametric satisfies Condition 1 of non-triviality: for every $S \subseteq V$ of size $t+1$ we know that there must be points $i,j \in S$ such that $d(i,j) = d(j,i) \geq t$ or in other words $x_{ij}^t = x_{ji}^t = 1$. Constraint 5 ensures that the ultrametric satisfies Condition 2 of non-triviality. To see this note that the constraint is active only when $\sum_{i,j \in S} x_{ij}^t = 0$ and $\sum_{i \in S, j \notin S}(1 - x_{ij}^t) = 0$. In other words $d(i,j) \leq t-1$ for every $i,j \in S$ and $S$ is a maximal such set since if $i \in S$ and $j \notin S$ then $d(i,j) \geq t$. Thus $S$ is an equivalence class under the relation $i \sim j$ iff $d(i,j) \leq t-1$ and so for every $i,j \in S$ we have $d(i,j) \leq |S| - 1$ or equivalently $x_{ij}^{|S|} = 0$. The ultrametric $d$ represented by a feasible solution $x_{ij}^t$ is given by $d(i,j) = \sum_{t=1}^{n-1} x_{ij}^t$.

**Definition 9.** *For any $\left\{ x_{ij}^t \mid t \in [n-1], i,j \in V \right\}$ let $E_t$ be defined as $E_t := \left\{ \{i,j\} \mid x_{ij}^t = 0 \right\}$. Note that if $x_{ij}^t$ is feasible for ILP-ultrametric then $E_t \subseteq E_{t+1}$ for any $t$ since $x_{ij}^t \geq x_{ij}^{t+1}$. The sets $\{E_t\}_{t=1}^{n-1}$ induce a natural sequence of graphs $\{G_t\}_{t=1}^{n-1}$ where $G_t = (V, E_t)$ with $V$ being the data set.*

For a fixed $t \in \{1, \ldots, n-1\}$ it is instructive to study the combinatorial properties of the so called *layer-$t$ problem*, where we restrict ourselves to the constraints corresponding to that particular $t$ and drop constraints (2) and (5) since they involve different layers in their expression.

$$\min \quad \sum_{\{i,j\} \in E(K_n)} \kappa(i,j) x_{ij}^t \tag{ILP-layer}$$

$$\text{s.t.} \quad x_{ij}^t + x_{jk}^t \geq x_{ik}^t \qquad \forall i,j,k \in V \tag{9}$$

$$\sum_{i,j \in S} x_{ij}^t \geq 2 \qquad \forall S \subseteq V, |S| = t+1 \tag{10}$$

$$x_{ij}^t = x_{ji}^t \qquad \forall i,j \in V \tag{11}$$

$$x_{ii}^t = 0 \qquad \forall i \in V \tag{12}$$

$$x_{ij}^t \in \{0,1\} \qquad \forall i,j \in V \tag{13}$$

The following lemma provides a combinatorial characterization of feasible solutions to the layer-$t$ problem.

**Lemma 10.** *Let $G_t = (V, E_t)$ be the graph as in Definition 9 corresponding to a solution $x_{ij}^t$ to the layer-$t$ problem ILP-layer. Then $G_t$ is a disjoint union of cliques of size $\leq t$. Moreover this exactly characterizes all feasible solutions of ILP-layer.*

*Proof.* We first note that $G_t = (V, E_t)$ must be a disjoint union of cliques since if $\{i,j\} \in E_t$ and $\{j,k\} \in E_t$ then $\{i,k\} \in E_t$ since $x_{ik}^t \leq x_{ij}^t + x_{jk}^t = 0$ due to constraint (9). Suppose there is a

clique in $G_t$ of size $> t$. Choose a subset $S$ of this clique of size $t + 1$. Then $\sum_{i,j \in S} x_{ij}^t = 0$ which violates constraint (10).

Conversely, let $E_t$ be a subset of edges such that $G_t = (V, E_t)$ is a disjoint union of cliques of size $\leq t$. Let $x_{ij}^t = 0$ if $\{i, j\} \in E_t$ and 1 otherwise. Clearly $x_{ij}^t = x_{ji}^t$ by definition. Suppose $x_{ij}^t$ violates constraint (9), so that there is a pair $i, j, k \in V$ such that $x_{ik}^t = 1$ but $x_{ij}^t = x_{jk}^t = 0$. However this implies that $G_t$ is not a disjoint union of cliques since $\{i, j\}, \{j, k\} \in E_t$ but $\{i, k\} \notin E_t$. Suppose $x_{ij}^t$ violates constraint (10) for some set $S$ of size $t + 1$. Therefore for every $i, j \in S$, we have $x_{ij}^t = 0$ since $x_{ij}^t = x_{ji}^t$ for every $i, j \in V$ and so $S$ must be a clique of size $t + 1$ in $G_t$ which is a contradiction. $\qquad\square$

By Lemma 10 the layer-$t$ problem is to find a subset $\overline{E}_t \subseteq E(K_n)$ of minimum weight under $\kappa$, such that the complement graph $G_t = (V, E_t)$ is a disjoint union of cliques of size $\leq t$. Note that this implies that the number of components in the complement graph is $\geq \lceil n/t \rceil$. The converse however, is not necessarily true: when $t = n - 1$ then the layer $t$-problem is the minimum (weighted) cut problem whose partitions may have size larger than 1. Our algorithmic approach is to solve an LP relaxation of ILP-ultrametric and then round the solution to obtain a feasible solution to ILP-ultrametric. The rounding however proceeds iteratively in a layer-wise manner and so we need to make sure that the rounded solution satisfies the inter-layer constraints (2) and (5). The following lemma gives a combinatorial characterization of solutions that satisfy these two constraints.

**Lemma 11.** *For every $t \in [n - 1]$, let $x_{ij}^t$ be feasible for the layer-$t$ problem ILP-layer. Let $G_t = (V, E_t)$ be the graph as in Definition 9 corresponding to $x_{ij}^t$, so that by Lemma 10, $G_t$ is a disjoint union of cliques $K_1^t, \ldots, K_{l_t}^t$ each of size at most $t$. Then $x_{ij}^t$ is feasible for ILP-ultrametric if and only if the following conditions hold.*

**Nested cliques** *For any $s \leq t$ every clique $K_p^s$ for some $p \in [l_s]$ in $G_s$ is a subclique of some clique $K_q^t$ in $G_t$ where $q \in [l_t]$.*

**Realization** *If $\left| K_p^t \right| = s$ for some $s \leq t$, then $G_s$ contains $K_p^t$ as a component clique, i.e., $K_q^s = K_p^t$ for some $q \in [l_s]$.*

*Proof.* Since $x_{ij}^t$ is feasible for the layer-$t$ problem ILP-layer it is feasible for ILP-ultrametric if and only if it satisfies constraints (2) and (5). The solution $x_{ij}^t$ satisfies constraint (2) if and only if $E_t \subseteq E_{t+1}$ by definition and so Condition Nested cliques follows.

Let us now assume that $x_{ij}^t$ is feasible for ILP-ultrametric, so that by the above argument Condition Nested cliques is satisfied. Note that every clique $K_p^t$ in the clique decomposition of $G_t$ corresponds to an equivalence class $S_t$ under the relation $i \sim j$ iff $x_{ij}^t = 0$. Moreover, by Lemma 10 we have $|S_t| \leq t$. Constraint (5) implies that $x_{ij}^{|S_t|} = 0$ for every $i, j \in S_t$. In other words, if $|S_t| = s \leq t$, then $x_{ij}^s = 0$ for every $i, j \in S_t$ and so $S_t$ is a subclique of some clique $K_q^s$ in the clique decomposition of $G_s$. However by Condition Nested cliques, $K_q^s$ must be a subclique of a clique $K_{p'}^t$ in the clique decomposition of $G_t$, since $s \leq t$. However, as $K_p^t \cap K_{p'}^t = S_t$ and the clique decomposition decomposes $G_t$ into a disjoint union of cliques, it follows that $S_t \subseteq K_q^s \subseteq K_{p'}^t = K_p^t = S_t$ and so $K_q^s = K_p^t$. Therefore Condition Realization is satisfied.

Conversely, suppose that $x_{ij}^t$ satisfies Conditions Nested cliques and Realization, so that by the argument in the paragraph above $x_{ij}^t$ satisfies constraint (2). Let us assume for the sake of contradiction that for a set $S \subseteq V$ and a $t \in [n - 1]$ constraint (5) is violated, i.e.,

$$\sum_{i,j \in S} x_{ij}^{|S|} > |S|^2 \left( \sum_{i,j \in S} x_{ij}^t + \sum_{\substack{i \in S \\ j \notin S}} \left(1 - x_{ij}^t\right) \right).$$

Since $x_{ij}^t \in \{0, 1\}$ it follows that $x_{ij}^t = 0$ for every $i, j \in S$ and $x_{ij}^t = 1$ for every $i \in S, j \notin S$ so that $S$ is a clique in $G_t$. Note that $|S| < t$ since $\sum_{i,j \in S} x_{ij}^{|S|} > 0$. This contradicts Condition Realization however, since $S$ is clearly not a clique in $G_{|S|}$. $\qquad\square$

The combinatorial interpretation of the individual layer-$t$ problems allow us to simplify the formulation of ILP-ultrametric by replacing the constraints for sets of a specific size (constraint (4)) by a global constraint about all sets (constraint (14)).

**Lemma 12.** *We may replace constraint* (4) *of* ILP-ultrametric *by the following equivalent constraint*

$$\sum_{j \in S} x_{ij}^t \geq |S| - t \qquad \forall t \in [n-1], S \subseteq V, i \in S. \tag{14}$$

*Proof.* Let $x_{ij}^t$ be a feasible solution to ILP-ultrametric. Note that if $|S| \leq t$ then the constraints are redundant since $x_{ij}^t \in \{0, 1\}$. Thus we may assume that $|S| > t$ and let $i$ be any vertex in $S$. Let us suppose for the sake of a contradiction that $\sum_{j \in S} x_{ij}^t < |S| - t$. This implies that there is a $t$ sized subset $S' \subseteq S \setminus \{i\}$ such that for every $j \in S'$ we have $x_{ij'}^t = 0$. In other words $\{i, j'\}$ is an edge in $G_t = (V, E_t)$ for every $j' \in S'$ and since $G_t$ is a disjoint union of cliques (constraint (3)), this implies the existence of a clique of size $t + 1$. Thus by Lemma 10, $x_{ij}^t$ could not have been a feasible solution to ILP-ultrametric.

Conversely, suppose $x_{ij}^t$ is feasible for the modified ILP where constraint (4) is replaced by constraint (14). Then again $G_t = (V, E_t)$ is a disjoint union of cliques since $x_{ij}^t$ satisfies constraint (3). Assume for contradiction that constraint (4) is violated: there is a set $S$ of size $t + 1$ such that $\sum_{i,j \in S} x_{ij}^t < 2$. Note that this implies that $\sum_{i,j} x_{ij}^t = 0$ since $x_{ij}^t = x_{ji}^t$ for every $i, j \in V$ and $t \in [n-1]$. Fix any $i \in S$, then $\sum_{j \in S} x_{ij}^t < 1 = |S| - t$ since $x_{ij}^t = x_{ji}^t$ by constraint (6), a violation of constraint (14). Thus $x_{ij}^t$ is feasible for ILP-ultrametric since it satisfies every other constraint by assumption. $\qquad \square$

## 4 Rounding an LP relaxation

In this section we consider the following natural LP relaxation for ILP-ultrametric. We keep the variables $x_{ij}^t$ for every $t \in [n-1]$ and $i, j \in V$ but relax the integrality constraint on the variables as well as drop constraint (5).

$$\min \quad \sum_{t=1}^{n-1} \sum_{\{i,j\} \in E(K_n)} \kappa(i,j) x_{ij}^t \qquad \text{(LP-ultrametric)}$$

$$\text{s.t.} \quad x_{ij}^t \geq x_{ij}^{t+1} \qquad \forall i, j \in V, t \in [n-2] \tag{15}$$

$$x_{ij}^t + x_{jk}^t \geq x_{ik}^t \qquad \forall i, j, k \in V, t \in [n-1] \tag{16}$$

$$\sum_{j \in S} x_{ij}^t \geq |S| - t \qquad \forall t \in [n-1], S \subseteq V, i \in S \tag{17}$$

$$x_{ij}^t = x_{ji}^t \qquad \forall i, j \in V, t \in [n-1] \tag{18}$$

$$x_{ii}^t = 0 \qquad \forall i, j \in V, t \in [n-1] \tag{19}$$

$$0 \leq x_{ij}^t \leq 1 \qquad \forall i, j \in V, t \in [n-1] \tag{20}$$

A feasible solution $x_{ij}^t$ to LP-ultrametric induces a sequence $\{d_t\}_{t \in [n-1]}$ of distance metrics over $V$ defined as $d_t(i,j) := x_{ij}^t$. Constraint 17 enforces an additional structure on this metric: informally points in a "large enough" subset $S$ should be spread apart according to the metric $d_t$. Metrics of type $d_t$ are called *spreading metrics* and were first studied in [19, 20] in relation to graph partitioning problems. The following lemma gives a technical interpretation of spreading metrics (see, e.g., [19, 20, 32]); we include a proof for completeness.

**Lemma 13.** *Let $x_{ij}^t$ be feasible for* LP-ultrametric *and for a fixed $t \in [n-1]$, let $d_t$ be the induced spreading metric. Let $i \in V$ be an arbitrary vertex and let $S \subseteq V$ be a set with $i \in S$ such that $|S| > (1 + \varepsilon)t$ for some $\varepsilon > 0$. Then $\max_{j \in S} d_t(i,j) > \frac{\varepsilon}{1+\varepsilon}$.*

*Proof.* For the sake of a contradiction suppose that for every $j \in S$ we have $d_t(i,j) = x^t_{ij} \leq \frac{\varepsilon}{1+\varepsilon}$. This implies that $x^t_{ij}$ violates constraint (17) leading to a contradiction:

$$\sum_{j \in S} x^t_{ij} \leq \frac{\varepsilon}{1+\varepsilon} |S| < |S| - t,$$

where the last inequality follows from $|S| > (1+\varepsilon)t$. $\qquad\square$

The following lemma shows that we can optimize over LP-ultrametric in polynomial time.

**Lemma 14.** *An optimal solution to LP-ultrametric can be computed in time polynomial in $n$ and* $\log (\max_{i,j} \kappa(i,j))$.

*Proof.* We argue in the standard fashion via the application of the Ellipsoid method (see e.g., [43]). As such it suffices to verify that the encoding length of the numbers is small (which is indeed the case here) and that the constraints can be separated in polynomial time in the size of the input, i.e., in $n$ and the logarithm of the absolute value of the largest coefficient. Since constraints of type (15), (16), (18), and (19) are polynomially many in $n$, we only need to check separation for constraints of type (17). Given a claimed solution $x^t_{ij}$ we can check constraint (17) by iterating over all $t \in [n-1]$, vertices $i \in V$, and sizes $m$ of the set $S$ from $t+1$ to $n$. For a fixed $t, i$, and set size $m$ sort the vertices in $V \setminus \{i\}$ in increasing order of distance from $i$ (according to the metric $d_t$) and let $\overline{S}$ be the first $m$ vertices in this ordering. If $\sum_{j \in \overline{S}} x^t_{ij} < m - t$ then clearly $x^t_{ij}$ is not feasible for LP-ultrametric, so we may assume that $\sum_{j \in \overline{S}} x^t_{ij} \geq m - t$. Moreover this is the only set to check: for any set $S \subseteq V$ containing $i$ such that $|S| = m$, $\sum_{j \in S} x^t_{ij} \geq \sum_{j \in \overline{S}} x^t_{ij} \geq m - t$. Thus for a fixed $t \in [n-1]$, $i \in V$ and set size $m$, it suffices to check that $x^t_{ij}$ satisfies constraint (17) for this subset $\overline{S}$. $\qquad\square$

From now on we will simply refer to a feasible solution to LP-ultrametric by the sequence of spreading metrics $\{d_t\}_{t \in [n-1]}$ it induces. The following definition introduces the notion of an open ball $\mathcal{B}_U (i, r, t)$ of radius $r$ centered at $i \in V$ according to the metric $d_t$ and restricted to the set $U \subseteq V$.

**Definition 15.** *Let $\{d_t \mid t \in [n-1]\}$ be the sequence of spreading metrics feasible for LP-ultrametric. Let $U \subseteq V$ be an arbitrary subset of $V$. For a vertex $i \in U$, $r \in \mathbb{R}$, and $t \in [n-1]$ we define the* open ball $\mathcal{B}_U (i, r, t)$ *of radius $r$ centered at $i$ as*

$$\mathcal{B}_U (i, r, t) \coloneqq \{j \in U \mid d_t(i,j) < r\} \subseteq U.$$

*If $U = V$ then we denote $\mathcal{B}_U (i, r, t)$ simply by $\mathcal{B} (i, r, t)$.*

**Remark 16.** *For every pair $i, j \in V$ we have $d_t(i,j) \geq d_{t+1}(i,j)$ by constraint (15). Thus for any subset $U \subseteq V$, $i \in U$, $r \in \mathbb{R}$, and $t \in [n-2]$, it holds $\mathcal{B}_U (i, r, t) \subseteq \mathcal{B}_U (i, r, t+1)$.*

To round LP-ultrametric to get a feasible solution for ILP-ultrametric, we will use the technique of *sphere growing* which was introduced in [33] to show an $O(\log n)$ approximation for the maximum multicommodity flow problem. Recall from Lemma 10 that a feasible solution to ILP-layer consists of a decomposition of the graph $G_t$ into a set of disjoint cliques of size at most $t$. One way to obtain such a decomposition is to choose an arbitrary vertex, grow a ball around this vertex until the expansion of this ball is below a certain threshold, chop off this ball and declare it as a partition and then recurse on the remaining vertices. This is the main idea behind sphere growing, and the parameters are chosen depending on the constraints of the specific problem (see, e.g., [25, 19, 14] for a few representative applications of this technique). The first step is to associate to every ball $\mathcal{B}_U (i, r, t)$ a volume $\mathrm{vol}\,(\mathcal{B}_U (i, r, t))$ and a boundary $\partial \mathcal{B}_U (i, r, t)$ so that its expansion is defined. For any $t \in [n-1]$ and $U \subseteq V$ we denote by $\gamma^U_t$ the value of the layer-$t$ objective for solution $d_t$ restricted to the set $U$, i.e.,

$$\gamma^U_t \coloneqq \sum_{\substack{i,j \in U \\ i < j}} \kappa(i,j) d_t(i,j).$$

When $U = V$ we refer to $\gamma^U_t$ simply by $\gamma_t$. Since $\kappa : V \times V \to \mathbb{R}_{\geq 0}$, it follows that $\gamma^U_t \leq \gamma_t$ for any $U \subseteq V$. We are now ready to define the volume, boundary, and expansion of a ball $\mathcal{B}_U (i, r, t)$. We use the definition of [19] modified for restrictions to arbitrary subsets $U \subseteq V$.

**Definition 17.** *[19] Let $U$ be an arbitrary subset of $V$. For a vertex $i \in U$, radius $r \in \mathbb{R}_{\geq 0}$, and $t \in [n-1]$, let $\mathcal{B}_U(i, r, t)$ be the ball of radius $r$ as in Definition 15. Then we define its* volume *as*

$$\mathrm{vol}\left(\mathcal{B}_U(i, r, t)\right) := \frac{\gamma_t^U}{n \log n} + \sum_{\substack{j, k \in \mathcal{B}_U(i,r,t) \\ j < k}} \kappa(j, k) d_t(j, k) + \sum_{\substack{j \in \mathcal{B}_U(i,r,t) \\ k \notin \mathcal{B}_U(i,r,t) \\ k \in U}} \kappa(j, k) \left(r - d_t(i, j)\right).$$

*The* boundary *of the ball $\partial \mathcal{B}_U(i, r, t)$ is the partial derivative of volume with respect to the radius:*

$$\partial \mathcal{B}_U(i, r, t) := \frac{\partial \, \mathrm{vol}\left(\mathcal{B}_U(i, r, t)\right)}{\partial r} = \sum_{\substack{j \in \mathcal{B}_U(i,r,t) \\ k \notin \mathcal{B}_U(i,r,t) \\ k \in U}} \kappa(j, k).$$

*The* expansion *$\phi(\mathcal{B}_U(i, r, t))$ of the ball $\mathcal{B}_U(i, r, t)$ is defined as the ratio of its boundary to its volume, i.e.,*

$$\phi\left(\mathcal{B}_U(i, r, t)\right) := \frac{\partial \mathcal{B}_U(i, r, t)}{\mathrm{vol}\left(\mathcal{B}_U(i, r, t)\right)}.$$

The following lemma shows that the volume of a ball $\mathcal{B}_U(i, r, t)$ is differentiable with respect to $r$ in the interval $(0, \Delta]$ except at finitely many points (see e.g., [19]).

**Lemma 18.** *Let $\mathcal{B}_U(i, r, t)$ be the ball corresponding to a set $U \subseteq V$, vertex $i \in U$, radius $r \in \mathbb{R}$ and $t \in [n-1]$. Then $\mathrm{vol}\left(\mathcal{B}_U(i, r, t)\right)$ is differentiable with respect to $r$ in the interval $(0, \Delta]$ except at finitely many points.*

*Proof.* Note that for any fixed $U \subseteq V$, $\mathrm{vol}\left(\mathcal{B}_U(i, r, t)\right)$ is a monotone non-decreasing function in $r$ since for a pair $j, k \in U$ such that $j \in \mathcal{B}_U(i, r, t)$ and $k \notin \mathcal{B}_U(i, r, t)$ we have $r - d_t(i, j) \leq d_t(j, k)$ otherwise $r - d_t(i, j) > d_t(j, k)$ so that $r > d_t(i, j) + d_t(j, k) \geq d_t(i, k)$, a contradiction to the fact that $k \notin \mathcal{B}_U(i, r, t)$. Therefore adding the vertex $k$ to the ball centered at $i$ is only going to increase its volume as $r - d_t(i, j) \leq d_t(j, k)$ (see Definition 15). Thus $\mathrm{vol}\left(\mathcal{B}_U(i, r, t)\right)$ is differentiable with respect to $r$ in the interval $(0, \Delta]$ except at finitely many points which correspond to a new vertex from $U$ being added to the ball. $\square$

The following theorem establishes that the rounding procedure of Algorithm 2 ensures that the cliques in $\mathcal{C}_t$ are "small" and that the cost of the edges removed to form them are not too high. It also shows that Algorithm 2 can be implemented to run in time polynomial in $n$.

**Theorem 19.** *Let $m_\varepsilon := \left\lfloor \frac{n-1}{1+\varepsilon} \right\rfloor$ as in Algorithm 2 and let $\left\{ x_{ij}^t \mid t \in [m_\varepsilon], i, j \in V \right\}$ be the output of Algorithm 2 run on a feasible solution $\{d_t\}_{t \in [n-1]}$ of LP-ultrametric and any choice of $\varepsilon \in (0, 1)$. For any $t \in [m_\varepsilon]$, we have that $x_{ij}^t$ is feasible for the layer-$\lfloor (1+\varepsilon) t \rfloor$ problem ILP-layer and there is a constant $c(\varepsilon) > 0$ depending only on $\varepsilon$ such that*

$$\sum_{\{i,j\} \in E(K_n)} \kappa(i, j) x_{ij}^t \leq c(\varepsilon) (\log n) \gamma_t.$$

*Moreover, Algorithm 2 can be implemented to run in time polynomial in $n$.*

*Proof.* We first show that for a fixed $t$, the constructed solution $x_{ij}^t$ is feasible for the layer-$\lfloor (1+\varepsilon) t \rfloor$ problem ILP-layer. Let $\mathcal{C}_t$ be as in Algorithm 2 so that $x_{ij}^t = 1$ if $i, j$ belong to different sets in $\mathcal{C}_t$ and $x_{ij}^t = 0$ otherwise. Let $G_t = (V, E_t)$ be as in Definition 9 corresponding to $x_{ij}^t$. Note that for any $t \in [m_\varepsilon]$, every $V_i \in \mathcal{C}_t$ is a clique in $G_t$ by construction (line 19) and for every distinct pair $V_i, V_j \in \mathcal{C}_t$ we have $V_i \cap V_j = \emptyset$ (lines 15 and 16). Therefore by Lemma 10, it suffices to prove that for any $V_i \in \mathcal{C}_t$, it holds $|V_i| \leq \lfloor (1+\varepsilon) t \rfloor$. If $V_i$ is added to $\mathcal{C}_t$ in line 9 then there is nothing to prove.

Thus let us assume that $V_i$ is of the form $\mathcal{B}_U(i, r, t)$ for some $U \subseteq V$ as in line 14 so that $\phi\left(\mathcal{B}_U(i, r, t)\right) \leq \frac{1}{\Delta} \log \left( \frac{\mathrm{vol}(\mathcal{B}_U(i, \Delta, t))}{\mathrm{vol}(\mathcal{B}_U(i, 0, t))} \right)$. Note that by Lemma 13 it suffices to show that there

**Algorithm 2:** Iterative rounding algorithm to find a low cost ultrametric

**Input**: Data set $V$, $\{d_t\}_{t\in[n-1]} : V \times V$, $\varepsilon > 0$, $\kappa : V \times V \to \mathbb{R}_{\geq 0}$

**Output**: A solution set of the form $\left\{ x_{ij}^t \in \{0,1\} \mid t \in \left[ \left\lfloor \frac{n-1}{1+\varepsilon} \right\rfloor \right], i, j \in V \right\}$

**1** $m_\varepsilon \leftarrow \left\lfloor \frac{n-1}{1+\varepsilon} \right\rfloor$

**2** $t \leftarrow m_\varepsilon$

**3** $\mathcal{C}_{t+1} \leftarrow \{V\}$

**4** $\Delta \leftarrow \frac{\varepsilon}{1+\varepsilon}$

**5** **while** $t \geq 1$ **do**

**6**     $\mathcal{C}_t \leftarrow \emptyset$

**7**     **for** $U \in \mathcal{C}_{t+1}$ **do**

**8**        **if** $|U| \leq (1+\varepsilon)t$ **then**

**9**           $\mathcal{C}_t \leftarrow \mathcal{C}_t \cup \{U\}$

**10**           Go to line 7

**11**        **end**

**12**        **while** $U \neq \emptyset$ **do**

**13**           Let $i$ be arbitrary in $U$

**14**           Let $r \in (0, \Delta]$ be s.t. $\phi\left(\mathcal{B}_U(i,r,t)\right) \leq \frac{1}{\Delta}\log\left(\frac{\mathrm{vol}(\mathcal{B}_U(i,\Delta,t))}{\mathrm{vol}(\mathcal{B}_U(i,0,t))}\right)$

**15**           $\mathcal{C}_t \leftarrow \mathcal{C}_t \cup \{\mathcal{B}_U(i,r,t)\}$

**16**           $U \leftarrow U \setminus \mathcal{B}_U(i,r,t)$

**17**        **end**

**18**     **end**

**19**     $x_{ij}^t = 1$ if $i \in U_1 \in \mathcal{C}_t$, $j \in U_2 \in C_t$ and $U_1 \neq U_2$, else $x_{ij}^t = 0$

**20**     $t \leftarrow t - 1$

**21** **end**

**22** **return** $\left\{ x_{ij}^t \mid t \in [m_\varepsilon], i, j \in V \right\}$

---

is such an $r \in (0, \Delta]$. This property follows from the rounding scheme due to [19] as we will explain now.

By Lemma 18 $\mathrm{vol}\left(\mathcal{B}_U(i,r,t)\right)$ is differentiable everywhere in the interval $(0, \Delta]$ except at finitely many points $X$. Let the set of discontinuous points be $X = \{x_1, x_2, \ldots, x_{k-1}\}$ with $x_0 = 0 < x_1 < x_2 \ldots x_{k-1} < x_k = \Delta$. We claim that there must be an $r \in (0, \Delta] \setminus X$ such that $\phi\left(\mathcal{B}_U(i,r,t)\right) \leq \frac{1}{\Delta}\log\left(\frac{\mathrm{vol}(\mathcal{B}_U(i,\Delta,t))}{\mathrm{vol}(\mathcal{B}_U(i,0,t))}\right)$. Let us assume for the sake of a contradiction that for every $r \in (0, \Delta] \setminus X$ we have $\phi\left(\mathcal{B}_U(i,r,t)\right) > \frac{1}{\Delta}\log\left(\frac{\mathrm{vol}(\mathcal{B}_U(i,\Delta,t))}{\mathrm{vol}(\mathcal{B}_U(i,0,t))}\right)$. However integrating both sides from 0 to $\Delta$ results in a contradiction:

$$\int_{r=0}^{\Delta} \phi\left(\mathcal{B}_U(i,r,t)\right) dr = \int_{r=0}^{\Delta} \frac{\partial \mathcal{B}_U(i,r,t)}{\mathrm{vol}\left(\mathcal{B}_U(i,r,t)\right)} dr \tag{21}$$

$$= \sum_{i=1}^{k} \int_{r=x_{i-1}}^{x_i} \frac{\partial \mathcal{B}_U(i,r,t)}{\mathrm{vol}\left(\mathcal{B}_U(i,r,t)\right)} dr \tag{22}$$

$$= \sum_{i=1}^{k} \int_{r=x_{i-1}}^{x_i} \frac{d\left(\mathrm{vol}\left(\mathcal{B}_U(i,r,t)\right)\right)}{\mathrm{vol}\left(\mathcal{B}_U(i,r,t)\right)} \tag{23}$$

$$\leq \log \mathrm{vol}\left(\mathcal{B}_U(i,\Delta,t)\right) - \log \mathrm{vol}\left(\mathcal{B}_U(i,0,t)\right) \tag{24}$$

$$= \int_{r=0}^{\Delta} \frac{1}{\Delta}\log\left(\frac{\mathrm{vol}\left(\mathcal{B}_U(i,\Delta,t)\right)}{\mathrm{vol}\left(\mathcal{B}_U(i,0,t)\right)}\right) dr, \tag{25}$$

where line 24 follows since $f$ is monotonic increasing. For any $t \in [m_\varepsilon]$ the set $\mathcal{C}_t$ is a disjoint partition of $V$ with balls of the form $\mathcal{B}_U(i,r,t')$ for some $t' \geq t$ and $U \subseteq U_l \in \mathcal{C}_{t'+1}$: this is easily seen by induction since $\mathcal{C}_{m_\varepsilon+1}$ is initialized as $V$. Further, a cluster $V_i$ is added to $\mathcal{C}_t$ either in line 15 in which case it is a ball of the form $\mathcal{B}_U(i,r,t)$ for some $U \in \mathcal{C}_{t+1}$, $i \in U$, and $r \in \mathbb{R}$ or it is added

in line 9 in which case it must have been a ball $\mathcal{B}_U\left(i',r',t'\right)$ for some $t' > t$, $U \subseteq U_l \in \mathcal{C}_{t'+1}$, $i' \in V$, and $r' \in \mathbb{R}$. Note that for any $t' \geq t$ and $U \subseteq V$, it holds $\gamma_{t'}^U \leq \gamma_t^U$ since for every pair $i,j \in V$ we have $\kappa(i,j) \geq 0$ and $d_t(i,j) \geq d_{t'}(i,j)$ because of constraint (15). Moreover, for any subset $U \subseteq V$ we have $\gamma_t^U \leq \gamma_t$ since $\kappa, d_t \geq 0$.

We claim that for any $t \in [m_\varepsilon]$ the total volume of the balls in $\mathcal{C}_t$ is at most $\left(2 + \frac{1}{\log n}\right)\gamma_t$. First note that the affine term $\frac{\gamma_{t'}^U}{n \log n}$ in the volume of a ball $\mathcal{B}_U\left(i,r,t'\right)$ in $\mathcal{C}_t$ is upper bounded by $\frac{\gamma_t}{n \log n}$ and appears at most $n$ times. Next we claim that the contribution to the total volume from the term involving the edges inside and crossing a ball $\mathcal{B}_U\left(i,r,t'\right) \in \mathcal{C}_t$ is at most $2\gamma_t$. This is because the balls are disjoint, $r - d_{t'}(i,k) \leq d_{t'}(j,k) \leq d_t(j,k)$ for the crossing edges of a ball $\mathcal{B}_U\left(i,r,t'\right) \in \mathcal{C}_t$ and a crossing edge contributes to the volume of at most 2 balls in $\mathcal{C}_t$. Note that for any $U \subseteq V$, $i \in U$, and $r \in \mathbb{R}_{\geq 0}$ we have $\mathrm{vol}\left(\mathcal{B}_U\left(i,r,t\right)\right) \in \left[\frac{\gamma_t^U}{n \log n}, \left(1 + \frac{1}{n \log n}\right)\gamma_t^U\right]$. Using this observation and the stopping condition of line 14 it follows that

$$
\begin{aligned}
\sum_{\{i,j\} \in E(K_n)} \kappa(i,j) x_{ij}^t &= \sum_{\substack{\{i,j\} \in E(K_n):\\ i,j \text{ separated in } \mathcal{C}_t}} \kappa(i,j)\\
&= \frac{1}{2} \underbrace{\sum_{\substack{\mathcal{B}_U\left(i,r,t'\right) \in \mathcal{C}_t:\\ t' \geq t\\ U \subseteq U_l \in \mathcal{C}_{t'+1}}} \sum_{\substack{j \in \mathcal{B}_U\left(i,r,t'\right)\\ k \notin \mathcal{B}_U\left(i,r,t'\right)}} \kappa(j,k)}_{\text{Since } \kappa \text{ is symmetric}}\\
&= \frac{1}{2} \sum_{\substack{\mathcal{B}_U\left(i,r,t'\right) \in \mathcal{C}_t:\\ t' \geq t\\ U \subseteq U_l \in \mathcal{C}_{t'+1}}} \partial \mathcal{B}_U\left(i,r,t'\right)\\
&= \frac{1}{2} \sum_{\substack{\mathcal{B}_U\left(i,r,t'\right) \in \mathcal{C}_t:\\ t' \geq t\\ U \subseteq U_l \in \mathcal{C}_{t'+1}}} \phi\left(\mathcal{B}_U\left(i,r,t'\right)\right) \mathrm{vol}\left(\mathcal{B}_U\left(i,r,t'\right)\right)\\
&\leq \sum_{\substack{\mathcal{B}_U\left(i,r,t'\right) \in \mathcal{C}_t:\\ t' \geq t\\ U \subseteq U_l \in \mathcal{C}_{t'+1}}} \frac{1}{2\Delta} \log\left(\frac{\mathrm{vol}\left(\mathcal{B}_U\left(i,\Delta,t'\right)\right)}{\mathrm{vol}\left(\mathcal{B}_U\left(i,0,t'\right)\right)}\right) \mathrm{vol}\left(\mathcal{B}_U\left(i,r,t'\right)\right)\\
&\leq \frac{1}{2\Delta} \underbrace{\left(\log\left(n \log n + 1\right)\right)}_{\text{via interval bounds}} \sum_{\substack{\mathcal{B}_U\left(i,r,t'\right) \in \mathcal{C}_t:\\ t' \geq t\\ U \subseteq U_l \in \mathcal{C}_{t'+1}}} \mathrm{vol}\left(\mathcal{B}_U\left(i,r,t'\right)\right)\\
&\leq \frac{1+\varepsilon}{2\varepsilon} \left(\log\left(n \log n + 1\right)\right) \underbrace{\left(2 + \frac{1}{\log n}\right)\gamma_t}_{\substack{\text{contribution of affine term} \leq \frac{\gamma_t}{\log n}\\ \text{contribution of edge terms} \leq 2\gamma_t}}\\
&\leq c(\varepsilon)(\log n)\gamma_t,
\end{aligned}
$$

for some constant $c(\varepsilon) > 0$ depending only on $\varepsilon$.

For the run time of Algorithm 2 note that the loop in line 5 runs for at most $n-1$ steps, while the loop in line 7 runs for at most $n$ steps. For a set $U \subseteq V$, to compute the ball $\mathcal{B}_U\left(i,r,t\right)$ of least radius $r$ such that $\phi\left(\mathcal{B}_U\left(i,r,t\right)\right) \leq \frac{1}{\Delta} \log\left(\frac{\mathrm{vol}(\mathcal{B}_U(i,\Delta,t))}{\mathrm{vol}(\mathcal{B}_U(i,0,t))}\right)$, sort the vertices in $U \setminus \{i\}$ in increasing order of distance from $i$ according to $d_t$. Let the vertices in $U \setminus \{i\}$ in this sorted order be $\{j_1, \ldots, j_{|U|-1}\}$. Then it suffices to check the expansion of the balls $\{i\}$ and $\{i\} \cup \{j_1, \ldots, j_k\}$ for every $k \in [|U|-1]$. It is straightforward to see that all the other steps in Algorithm 2 run in time polynomial in $n$. $\qquad\square$

**Remark 20.** *A discrete version of the volumetric argument for region growing can be found in [26].*

We are now ready to prove the main theorem showing that we can obtain a low cost non-trivial ultrametric from Algorithm 2.

**Theorem 21.** *Let $\{x_{ij}^t \mid t \in [m_\varepsilon], i, j \in V\}$ be the output of Algorithm 2 on an optimal solution $\{d_t\}_{t \in [n-1]}$ of LP-ultrametric for any choice of $\varepsilon \in (0,1)$. Define the sequence $\{y_{ij}^t\}$ for every $t \in [n-1]$ and $i, j \in V$ as*

$$y_{ij}^t := \begin{cases} x_{ij}^{\lfloor t/(1+\varepsilon) \rfloor} & \text{if } t > 1 + \varepsilon \\ 1 & \text{if } t \le 1 + \varepsilon. \end{cases}$$

*Then $y_{ij}^t$ is feasible for ILP-ultrametric and satisfies*

$$\sum_{t=1}^{n-1} \sum_{\{i,j\} \in E(K_n)} \kappa(i,j) y_{ij}^t \le (2c(\varepsilon) \log n) \, \text{OPT}$$

*where* OPT *is the optimal solution to ILP-ultrametric and $c(\varepsilon)$ is the constant in the statement of Theorem 21.*

*Proof.* Note that by Theorem 19 for every $t \in [m_\varepsilon]$, $x_{ij}^t$ is feasible for the layer-$\lfloor (1 + \varepsilon)t \rfloor$ problem ILP-layer and that there is a constant $c(\varepsilon) > 0$ such that for every $t \in [m_\varepsilon]$, we have $\sum_{\{i,j\} \in E(K_n)} \kappa(i,j) x_{ij}^t \le (c(\varepsilon) \log n) \gamma_t$.

Let $y_{ij}^t$ be as in the statement of the theorem. The graph $G_t = (V, E_t)$ as in Definition 9 corresponding to $y_{ij}^t$ for $t \le 1 + \varepsilon$ consists of isolated vertices, i.e., cliques of size 1: By definition $y_{ij}^t$ is feasible for the layer-$t$ problem ILP-layer. The collection $\mathcal{C}_1$ corresponding to $x_{ij}^1$ consists of cliques of size at most $1 + \varepsilon$, however since $0 < \varepsilon < 1$ it follows that the cliques in $\mathcal{C}_1$ are isolated vertices and so $x_{ij}^1 = 1$ for every $\{i, j\} \in E(K_n)$. Thus $\sum_{i,j} \kappa(i,j) y_{ij}^t = \sum_{i,j} \kappa(i,j) x_{ij}^1 \le (c(\varepsilon) \log n) \gamma_1$ for $t \le 1 + \varepsilon$ by Theorem 19. Moreover for every $t > 1 + \varepsilon$, we have $\sum_{i,j} \kappa(i,j) y_{ij}^t \le (c(\varepsilon) \log n) \gamma_{\lfloor t/(1+\varepsilon) \rfloor}$ again by Theorem 19. We claim that $y_{ij}^t$ is feasible for ILP-ultrametric. The solution $y_{ij}^t$ corresponds to the collection $\mathcal{C}_{\lfloor \frac{t}{1+\varepsilon} \rfloor}$ for $t > 1 + \varepsilon$ or to the collection $\mathcal{C}_1$ for $t \le 1 + \varepsilon$ from Algorithm 2. For any $t < m_\varepsilon$, every ball $\mathcal{B}_U(i, r, t) \in \mathcal{C}_t$ comes from the refinement of a ball $\mathcal{B}_{U'}(i', r', t')$ for some $i' \in V$, $r' \ge r$, $t' \ge t$ and $U' \supseteq U$. Thus $y_{ij}^t$ satisfies Condition Nested cliques of Lemma 11. On the other hand line 8 ensures that if $|\mathcal{B}_U(i, r, t)| = \lfloor (1 + \varepsilon)s \rfloor$ for some $U \subseteq V$ and $s < t$ then $\mathcal{B}_U(i, r, t)$ also appears as a ball in $\mathcal{C}_s$. Therefore $y_{ij}^t$ also satisfies Condition Realization of Lemma 11 and so is feasible for ILP-ultrametric. The cost of $y_{ij}^t$ is at most

$$\sum_{t=1}^{n-1} \sum_{\{i,j\} \in E(K_n)} \kappa(i,j) y_{ij}^t \le (c(\varepsilon) \log n) \left( \gamma_1 + \sum_{t=2}^{n-1} \gamma_{\lfloor t/(1+\varepsilon) \rfloor} \right)$$

$$\le 2c(\varepsilon) \log n \sum_{t=1}^{n-1} \gamma_t$$

$$\le 2c(\varepsilon) \log n \, \text{OPT},$$

where we use the fact that $\sum_{t=1}^{n-1} \gamma_t = \text{OPT}(LP) \le \text{OPT}$ since LP-ultrametric is a relaxation of ILP-ultrametric. $\qquad\square$

Theorem 21 implies the following corollary where we put everything together to obtain a hierarchical clustering of $V$ in time polynomial in $n$ with $|V| = n$. Let $\mathcal{T}$ denote the set of all possible hierarchical clusterings of $V$.

**Corollary 22.** *Given a data set $V$ of $n$ points and a similarity function $\kappa : V \times V \to \mathbb{R}_{\ge 0}$, Algorithm 3 returns a hierarchical clustering $T$ of $V$ satisfying*

$$\text{cost}(T) \le O(\log n) \min_{T' \in \mathcal{T}} \text{cost}(T').$$

*Moreover Algorithm 3 runs in time polynomial in $n$ and $\log (\max_{i,j \in V} \kappa(i,j))$.*

---

**Algorithm 3:** Hierarchical clustering of $V$ for cost function (1)

---

**Input**: Data set $V$ of $n$ points, similarity function $\kappa : V \times V \to \mathbb{R}_{\geq 0}$
**Output**: Hierarchical clustering of $V$

1 Solve LP-ultrametric to obtain optimal sequence of spreading metrics $\{d_t \mid d_t : V \times V \to [0,1]\}$
2 Fix a choice of $\varepsilon \in (0,1)$
3 $m_\varepsilon \leftarrow \left\lfloor \frac{n-1}{1+\varepsilon} \right\rfloor$
4 Let $\left\{ x_{ij}^t \mid t \in [m_\varepsilon] \right\}$ be the output of Algorithm 2 on $V, \kappa, \{d_t\}_{t \in [n-1]}$
5 Let $y_{ij}^t := \begin{cases} x_{ij}^{\lfloor t/(1+\varepsilon) \rfloor} & \text{if } t > 1 + \varepsilon \\ 1 & \text{if } t \leq 1 + \varepsilon \end{cases}$ for every $t \in [n-1], i, j \in E(K_n)$
6 $d(i,j) \leftarrow \sum_{t=1}^{n-1} y_{ij}^t$ for every $i, j \in E(K_n)$
7 $d(i,i) \leftarrow 0$ for every $i \in V$
8 Let $r, T$ be the output of Algorithm 1 on $V, d$
9 **return** $r, T$

---

*Proof.* Let $\widehat{T}$ be the optimal hierarchical clustering according to cost function (1). By Corollary 8 and Theorem 21 we can find a hierarchical clustering $T$ satisfying

$$\sum_{\{i,j\} \in E(K_n)} \kappa(i,j)(|\text{leaves}(T[\text{lca}(i,j)])| - 1) \leq O(\log n) \left( \sum_{\{i,j\} \in E(K_n)} \kappa(i,j) \left( \left| \text{leaves}(\widehat{T}[\text{lca}(i,j)]) \right| - 1 \right) \right).$$

Let $K := \sum_{\{i,j\} \in E(K_n)} \kappa(i,j)$. Then it follows from the above expression that $\text{cost}(T) \leq O(\log n) \text{cost}(\widehat{T}) - O(\log n)K + K \leq O(\log n) \text{cost}(\widehat{T})$.

We can find an optimal solution to LP-ultrametric due to Lemma 14 using the Ellipsoid algorithm in time polynomial in $n$ and $\log(\max_{i,j \in V} \kappa(i,j))$. Algorithm 2 runs in time polynomial in $n$ due to Theorem 19. Finally, Algorithm 1 runs in time $O(n^3)$ due to Lemma 7. $\qquad\square$

## 5 Generalized Cost Function

In this section we study the following natural generalization of cost function (1) also introduced by [16] where the distance between the two points is scaled by a function $f : \mathbb{R}_{\geq 0} \to \mathbb{R}_{\geq 0}$, i.e.,

$$\text{cost}_f(T) := \sum_{\{i,j\} \in E(K_n)} \kappa(i,j) f\left(|\text{leaves}\, T[\text{lca}(i,j)]|\right). \tag{26}$$

In order that cost function (26) makes sense, $f$ should be strictly increasing and satisfy $f(0) = 0$. Possible choices for $f$ could be $\{x^2, e^x - 1, \log(1+x)\}$. The top-down heuristic in [16] finds the optimal hierarchical clustering up to an approximation factor of $c_n \log n$ with $c_n$ being defined as

$$c_n := 3\alpha_n \max_{1 \leq n' \leq n} \frac{f(n')}{f(\lceil n'/3 \rceil)}$$

and where $\alpha_n$ is the approximation factor from the Sparsest Cut algorithm used.

A naive approach to solving this problem using the ideas of Algorithm 2 would be to replace the objective function of ILP-ultrametric by

$$\sum_{\{i,j\} \in E(K_n)} \kappa(i,j) f\left(\sum_{t=1}^{n-1} x_{ij}^t\right).$$

This makes the corresponding analogue of LP-ultrametric non-linear however, and for a general $\kappa$ and $f$ it is not clear how to compute an optimum solution in polynomial time. One possible solution is to assume that $f$ is convex and use the Frank-Wolfe algorithm to compute an optimum solution. That still leaves the problem of how to relate $f\left(\sum_{t=1}^{n-1} x_{ij}^t\right)$ to $\sum_{t=1}^{n-1} f\left(x_{ij}^t\right)$ as one would have to do to get a corresponding version of Theorem 21. The following simple observation provides an alternate way of tackling this problem.

**Observation 23.** *Let $d : V \times V \to \mathbb{R}$ be an ultrametric and $f : \mathbb{R}_{\geq 0} \to \mathbb{R}_{\geq 0}$ be a strictly increasing function such that $f(0) = 0$. Define the function $f(d) : V \times V \to \mathbb{R}$ as $f(d)(i,j) := f(d(i,j))$. Then $f(d)$ is also an ultrametric on $V$.*

Therefore by Corollary 8 to find a minimum cost hierarchical clustering $T$ of $V$ according to the cost function (26), it suffices to minimize $\langle \kappa, d \rangle$ where $d$ is the $f$-image of a non-trivial ultrametric as in Definition 2. The following lemma lays down the analogue of Conditions 1 and 2 from Definition 2 that the $f$-image of a non-trivial ultrametric satisfies.

**Lemma 24.** *Let $f : \mathbb{R}_{\geq 0} \to \mathbb{R}_{\geq 0}$ be a strictly increasing function satisfying $f(0) = 0$. An ultrametric $d$ on $V$ is the $f$-image of a non-trivial ultrametric on $V$ iff*

1. *for every non-empty set $S \subseteq V$, there is a pair of points $i, j \in S$ such that $d(i,j) \geq f(|S| - 1)$,*

2. *for any $t$ if $S_t$ is an equivalence class of $V$ under the relation $i \sim j$ iff $d(i,j) \leq t$, then $\max_{i,j \in S_t} d(i,j) \leq f(|S_t| - 1)$.*

*Proof.* If $d$ is the $f$-image of a non-trivial ultrametric $d'$ on $V$ then clearly $d$ satisfies Conditions 1 and 2. Conversely, let $d$ be an ultrametric on $V$ satisfying Conditions 1 and 2. Note that $f$ is strictly increasing and $V$ is a finite set and thus $f^{-1}$ exists and is strictly increasing as well, with $f^{-1}(0) = 0$. Define $d'$ as $d'(i,j) := f^{-1}(d(i,j))$ for every $i, j \in V$. By Observation 23 $d'$ is an ultrametric on $V$ satisfying Conditions 1 and 2 of Definition 2 and so $d'$ is a non-trivial ultrametric on $V$. $\qquad \square$

Lemma 24 allows us to write the analogue of ILP-ultrametric for finding the minimum cost ultrametric that is the $f$-image of a non-trivial ultrametric on $V$. Note that by Lemma 4 the range of such an ultrametric is the set $\{f(0), f(1), \ldots, f(n-1)\}$. We have the binary variables $x_{ij}^t$ for every distinct pair $i, j \in V$ and $t \in [n-1]$, where $x_{ij}^t = 1$ if $d(i,j) \geq f(t)$ and $x_{ij}^t = 0$ if $d(i,j) < f(t)$.

$$\min \quad \sum_{t=1}^{n-1} \sum_{\{i,j\} \in E(K_n)} \kappa(i,j) \left( f(t) - f(t-1) \right) x_{ij}^t \qquad \text{(f-ILP-ultrametric)}$$

$$\text{s.t.} \quad x_{ij}^t \geq x_{ij}^{t+1} \qquad \forall i, j \in V, t \in [n-2] \qquad (27)$$

$$x_{ij}^t + x_{jk}^t \geq x_{ik}^t \qquad \forall i, j, k \in V, t \in [n-1] \qquad (28)$$

$$\sum_{i,j \in S} x_{ij}^t \geq 2 \qquad \forall t \in [n-1], S \subseteq V, |S| = t+1 \qquad (29)$$

$$\sum_{i,j \in S} x_{ij}^{|S|} \leq |S|^2 \left( \sum_{i,j \in S} x_{ij}^t + \sum_{\substack{i \in S \\ j \notin S}} \left( 1 - x_{ij}^t \right) \right) \forall t \in [n-1], S \subseteq V \qquad (30)$$

$$x_{ij}^t = x_{ji}^t \qquad \forall i, j \in V, t \in [n-1] \qquad (31)$$

$$x_{ii}^t = 0 \qquad \forall i \in V, t \in [n-1] \qquad (32)$$

$$x_{ij}^t \in \{0,1\} \qquad \forall i, j \in V, t \in [n-1] \qquad (33)$$

If $x_{ij}^t$ is a feasible solution to f-ILP-ultrametric then the ultrametric represented by it is defined as

$$d(i,j) := \sum_{t=1}^{n-1} (f(t) - f(t-1)) x_{ij}^t.$$

Constraint (29) ensures that $d$ satisfies Condition 1 of Lemma 24, since for every $S \subseteq V$ of size $t + 1$ we have a pair $i, j \in S$ such that $d(i,j) \geq f(t)$. Similarly constraint (30) ensures that $d$ satisfies Condition 2 of Lemma 24 since it is active if and only if $S$ is an equivalence class of $V$ under the relation $i \sim j$ iff $d(i,j) < f(t)$. In this case Condition 2 of Lemma 24 requires $\max_{i,j \in S} d(i,j) \leq f(|S| - 1)$ or in other words $x_{ij}^{|S|} = 0$ for every $i, j \in S$.

Similar to ILP-layer we define an analogous *layer-t problem* where we fix a choice of $t \in [n-1]$ and drop the constraints that relate the different layers to each other.

$$\min \sum_{\{i,j\} \in E(K_n)} \kappa(i,j)\left(f(t)-f(t-1)\right) x_{ij}^t \qquad \text{(f-ILP-layer)}$$

$$\text{s.t.} \quad x_{ij}^t + x_{jk}^t \geq x_{ik}^t \qquad \forall i,j,k \in V \tag{34}$$

$$\sum_{i,j \in S} x_{ij}^t \geq 2 \qquad \forall S \subseteq V, |S| = t+1 \tag{35}$$

$$x_{ij}^t = x_{ji}^t \qquad \forall i,j \in V \tag{36}$$

$$x_{ii}^t = 0 \qquad \forall i \in V \tag{37}$$

$$x_{ij}^t \in \{0,1\} \qquad \forall i,j \in V \tag{38}$$

Note that f-ILP-ultrametric and f-ILP-layer differ from ILP-ultrametric and ILP-layer respectively only in the objective function. Therefore Lemmas 10 and 11 also give a combinatorial characterization of the set of feasible solutions to f-ILP-layer and f-ILP-ultrametric respectively.

Similarly by Lemma 12 we may replace constraint (29) by the following equivalent constraint over all subsets of $V$

$$\sum_{j \in S} x_{ij}^t \geq |S| - t \qquad \forall t \in [n-1], S \subseteq V, i \in S.$$

This provides the analogue of LP-ultrametric in which we drop constraint (30) and enforce it in the rounding procedure.

$$\min \sum_{t=1}^{n-1} \sum_{\{i,j\} \in E(K_n)} \kappa(i,j)\left(f(t)-f(t-1)\right) x_{ij}^t \qquad \text{(f-LP-ultrametric)}$$

$$\text{s.t.} \quad x_{ij}^t \geq x_{ij}^{t+1} \qquad \forall i,j \in V, t \in [n-2] \tag{39}$$

$$x_{ij}^t + x_{jk}^t \geq x_{ik}^t \qquad \forall i,j,k \in V, t \in [n-1] \tag{40}$$

$$\sum_{j \in S} x_{ij}^t \geq |S| - t \qquad \forall t \in [n-1], S \subseteq V, i \in S \tag{41}$$

$$x_{ij}^t = x_{ji}^t \qquad \forall i,j \in V, t \in [n-1] \tag{42}$$

$$x_{ii}^t = 0 \qquad \forall i \in V, t \in [n-1] \tag{43}$$

$$0 \leq x_{ij}^t \leq 1 \qquad \forall i,j \in V, t \in [n-1] \tag{44}$$

Since f-LP-ultrametric differs from LP-ultrametric only in the objective function, it follows from Lemma 14 that an optimum solution to f-LP-ultrametric can be computed in time polynomial in $n$. As before, a feasible solution $x_{ij}^t$ of f-LP-ultrametric induces a sequence $\{d_t\}_{t \in [n-1]}$ of spreading metrics on $V$ defined as $d_t(i,j) := x_{ij}^t$. Note that in contrast to the ultrametric $d$, the spreading metrics $\{d_t\}_{t \in [n-1]}$ are independent of the function $f$.

Let $\mathcal{B}_U(i,r,t)$ be a ball of radius $r$ centered at $i \in U$ for some set $U \subseteq V$ as in Definition 15. For a subset $U \subseteq V$, let $\gamma_t^U$ be defined as before to be the value of the layer-$t$ objective corresponding to a solution $d_t$ of f-LP-ultrametric restricted to $U$, i.e.,

$$\gamma_t^U := \sum_{\substack{i,j \in U \\ i<j}} \left(f(t)-f(t-1)\right) \kappa(i,j) d_t(i,j).$$

As before, we denote $\gamma_t^V$ by $\gamma_t$. We will associate a volume $\text{vol}\left(\mathcal{B}_U(i,r,t)\right)$ and a boundary $\partial\mathcal{B}_U(i,r,t)$ to the ball $\mathcal{B}_U(i,r,t)$ as in Section 4.

**Definition 25.** *Let $U$ be an arbitrary subset of $V$. For a vertex $i \in U$, radius $r \in \mathbb{R}_{\geq 0}$, and $t \in [n-1]$, let $\mathcal{B}_U(i, r, t)$ be the ball of radius $r$ as in Definition 15. Then we define its* volume *as*

$$\mathrm{vol}\left(\mathcal{B}_U(i, r, t)\right) := \frac{\gamma_t^U}{n \log n} + (f(t) - f(t-1)) \left( \sum_{\substack{j,k \in \mathcal{B}_U(i,r,t) \\ j<k}} \kappa(j,k) d_t(j,k) + \sum_{\substack{j \in \mathcal{B}_U(i,r,t) \\ k \notin \mathcal{B}_U(i,r,t) \\ k \in U}} \kappa(j,k)\left(r - d_t(i,j)\right) \right).$$

*The* boundary *of the ball $\partial \mathcal{B}_U(i, r, t)$ is the partial derivative of volume with respect to the radius:*

$$\partial \mathcal{B}_U(i, r, t) := (f(t) - f(t-1)) \left( \frac{\partial\, \mathrm{vol}\left(\mathcal{B}_U(i, r, t)\right)}{\partial r} \right) = (f(t) - f(t-1)) \left( \sum_{\substack{j \in \mathcal{B}_U(i,r,t) \\ k \notin \mathcal{B}_U(i,r,t) \\ k \in U}} \kappa(j,k) \right).$$

*The* expansion $\phi\left(\mathcal{B}_U(i, r, t)\right)$ *of the ball $\mathcal{B}_U(i, r, t)$ is defined as the ratio of its boundary to its volume, i.e.,*

$$\phi\left(\mathcal{B}_U(i, r, t)\right) := \frac{\partial \mathcal{B}_U(i, r, t)}{\mathrm{vol}\left(\mathcal{B}_U(i, r, t)\right)}.$$

Note that the expansion $\phi\left(\mathcal{B}_U(i, r, t)\right)$ of Definition 25 is the same as in Definition 17 since the $(f(t) - f(t-1))$ term cancels out. Thus one could run Algorithm 2 with the same notion of volume as in Definition 17, however in that case the analogous versions of Theorems 19 and 21 do not follow as naturally. The following is then a simple corollary of Theorem 19.

**Corollary 26.** *Let $m_\varepsilon := \left\lfloor \frac{n-1}{1+\varepsilon} \right\rfloor$ as in Algorithm 2. Let $\left\{ x_{ij}^t \mid t \in [n-1], i, j \in V \right\}$ be the output of Algorithm 2 using the notion of volume, boundary and expansion as in Definition 25, on a feasible solution to f-LP-ultrametric and any choice of $\varepsilon \in (0, 1)$. For any $t \in [m_\varepsilon]$, we have that $x_{ij}^t$ is feasible for the layer-$\lfloor (1+\varepsilon)t \rfloor$ problem f-ILP-layer and there is a constant $c(\varepsilon) > 0$ depending only on $\varepsilon$ such that*

$$\sum_{\{i,j\} \in E(K_n)} \kappa(i,j)\left(f(t) - f(t-1)\right) x_{ij}^t \leq (c(\varepsilon) \log n)\, \gamma_t.$$

Corollary 26 allows us to prove the analogue of Theorem 21, i.e., we can use Algorithm 2 to get an ultrametric that is an $f$-image of a non-trivial ultrametric and whose cost is at most $O(\log n)$ times the cost of an optimal hierarchical clustering according to cost function (26).

**Theorem 27.** *Let $\left\{ x_{ij}^t \mid t \in [m_\varepsilon], i, j \in V \right\}$ be the output of Algorithm 2 using the notion of volume, boundary, and expansion as in Definition 25 on an optimal solution $\{d_t\}_{t \in [n-1]}$ of f-LP-ultrametric for any choice of $\varepsilon \in (0, 1)$. Define the sequence $\left\{ y_{ij}^t \right\}$ for every $t \in [n-1]$ and $i, j \in V$ as*

$$y_{ij}^t := \begin{cases} x_{ij}^{\lfloor t/(1+\varepsilon) \rfloor} & \text{if } t > 1 + \varepsilon \\ 1 & \text{if } t \leq 1 + \varepsilon. \end{cases}$$

*Then $y_{ij}^t$ is feasible for f-ILP-ultrametric and there is a constant $c(\varepsilon) > 0$ such that*

$$\sum_{t=1}^{n-1} \sum_{\{i,j\} \in E(K_n)} \kappa(i,j)\left(f(t) - f(t-1)\right) y_{ij}^t \leq (c(\varepsilon) \log n)\, \mathrm{OPT}$$

*where* OPT *is the optimal solution to f-ILP-ultrametric.*

*Proof.* Immediate from Corollary 26 and Theorem 21. □

Finally we put everything together to obtain the corresponding Algorithm 4 that outputs a hierarchical clustering of $V$ of cost at most $O\left(\log n\right)$ times the optimal clustering according to cost function (26).

**Corollary 28.** *Given a data set $V$ of $n$ points and a similarity function $\kappa : V \times V \to \mathbb{R}$, Algorithm 4 returns a hierarchical clustering $T$ of $V$ satisfying*

$$\text{cost}_f(T) \leq O\left(a_n + \log n\right) \min_{T' \in \mathcal{T}} \text{cost}_f(T'),$$

*where $a_n := \max_{n' \in [n]} f(n') - f(n'-1)$. Moreover Algorithm 4 runs in time polynomial in $n$, $\log f(n)$ and $\log\left(\max_{i,j \in V} \kappa(i,j)\right)$.*

*Proof.* Let $\widehat{T}$ be an optimal hierarchical clustering according to cost function (26). By Corollary 8, Lemma 24 and Theorem 27 it follows that we can find a hierarchical clustering $T$ satisfying

$$\sum_{\{i,j\} \in E(K_n)} \kappa(i,j) f\left(|\text{leaves}(T[\text{lca}(i,j)]| - 1\right) \leq O(\log n) \left( \sum_{\{i,j\} \in E(K_n)} \kappa(i,j) f\left(\left|\text{leaves}(\widehat{T}[\text{lca}(i,j)]\right| - 1\right) \right).$$

Recall that $\text{cost}_f(T) := \sum_{\{i,j\} \in E(K_n)} \kappa(i,j) f\left(|\text{leaves}(T[\text{lca}(i,j)]|\right)$. Let $K := \sum_{\{i,j\} \in E(K_n)} \kappa(i,j)$. Note that for any hierarchical clustering $T'$ we have $K \leq \text{cost}_f(T')$ since $f$ is an increasing function. From the above expression we infer that

$$\text{cost}_f(T) - a_n K \leq \sum_{\{i,j\} \in E(K_n)} \kappa(i,j) f\left(|\text{leaves}(T[\text{lca}(i,j)]| - 1\right) \leq O(\log n) \text{cost}_f(\widehat{T}),$$

and so $\text{cost}_f(T) \leq O(\log n) \text{cost}_f(\widehat{T}) + a_n K \leq O(a_n + \log n) \text{cost}_f(\widehat{T})$. We can find an optimal solution to f-LP-ultrametric due to Lemma 14 using the Ellipsoid algorithm in time polynomial in $n$, $\log f(n)$, and $\log\left(\max_{i,j \in V} \kappa(i,j)\right)$. Note the additional $\log f(n)$ in the running time since now we need to binary search over the interval $[0, \max_{i,j \in V} \kappa(i,j) \cdot f(n) \cdot n]$. Algorithm 2 runs in time polynomial in $n$ due to Theorem 19. Finally, Algorithm 1 runs in time $O(n^3)$ due to Lemma 7. $\square$

---

**Algorithm 4:** Hierarchical clustering of $V$ for cost function (26)

**Input**: Data set $V$ of $n$ points, similarity function $\kappa : V \times V \to \mathbb{R}_{\geq 0}$, $f : \mathbb{R}_{\geq 0} \to \mathbb{R}_{\geq 0}$ strictly increasing with $f(0) = 0$

**Output**: Hierarchical clustering of $V$

1 Solve f-LP-ultrametric to obtain optimal sequence of spreading metrics $\{d_t \mid d_t : V \times V \to [0,1]\}$
2 Fix a choice of $\varepsilon \in (0,1)$
3 $m_\varepsilon \leftarrow \left\lfloor \frac{n-1}{1+\varepsilon} \right\rfloor$
4 Let $\left\{x_{ij}^t \mid t \in [m_\varepsilon]\right\}$ be the output of Algorithm 2 on $V, \kappa, \{d_t\}_{t \in [n-1]}$
5 Let $y_{ij}^t := \begin{cases} x_{ij}^{\lfloor t/(1+\varepsilon) \rfloor} & \text{if } t > 1 + \varepsilon \\ 1 & \text{if } t \leq 1 + \varepsilon \end{cases}$ for every $t \in [n-1], i,j \in E(K_n)$
6 $d(i,j) \leftarrow \sum_{t=1}^{n-1} \left(f(t) - f(t-1)\right) y_{ij}^t$ for every $i,j \in E(K_n)$
7 $d(i,i) \leftarrow 0$ for every $i \in V$
8 Let $r, T$ be the output of Algorithm 1 on $V, f^{-1}(d)$
9 **return** $r, T$

---

## 6 Experiments

Finally, we describe the experiments we performed. For small data sets ILP-ultrametric and f-ILP-ultrametric describe integer programming formulations that allow us to compute the exact optimal hierarchical clustering for cost functions (1) and (26) respectively. We implement f-ILP-ultrametric where one can plug in any strictly increasing function $f$ satisfying $f(0) = 0$. In particular, setting $f(x) = x$ gives us ILP-ultrametric. We use the Mixed Integer Programming (MIP) solver Gurobi 6.5 [27]. Similarly, we also implement Algorithms 1, 2, and 4 using Gurobi as our LP solver. Note that Algorithm 4 needs to fix a parameter choice $\varepsilon \in (0,1)$. In Sections 4 and 5 we did not discuss the effect of the choice of the parameter $\varepsilon$ in detail. In particular, we need to choose an $\varepsilon$ small enough such that for every $U \subseteq V$ encountered in Algorithm 2, $\text{vol}\left(\mathcal{B}_U(i, \Delta, t)\right)$ is of the same sign as

$\mathrm{vol}\left(\mathcal{B}_U\left(i, 0, t\right)\right)$ for every $t \in [n-1]$, so that $\log\left(\frac{\mathrm{vol}(\mathcal{B}_U(i,\Delta,t))}{\mathrm{vol}(\mathcal{B}_U(i,0,t))}\right)$ is defined. In our experiments we start with a particular value of $\varepsilon$ (say 0.5) and halve it till the volumes have the same sign. For the sake of exposition, we limit ourselves to the following choices for the function $f$

$$\left\{x, x^2, \log(1+x), e^x - 1\right\}.$$

By Lemma 14 we can optimize over f-LP-ultrametric in time polynomial in $n$ using the Ellipsoid method. In practice however, we use the *dual simplex* method where we separate triangle inequality constraints (40) and spreading constraints (41) to obtain fast computations. For the similarity function $\kappa : V \times V \to \mathbb{R}$ we limit ourselves to using *cosine similarity* and the *Gaussian kernel* with $\sigma = 1$. They are defined formally below.

**Definition 29** (Cosine similarity)**.** *Given a data set $V \in \mathbb{R}^m$ for some $m \geq 0$, the cosine similarity $\kappa_{cos}$ is defined as $\kappa_{cos}(x, y) \coloneqq \frac{\langle x, y\rangle}{\|x\|\|y\|}$.*

Since the LP rounding Algorithm 2 assumes that $\kappa \geq 0$ in practice we implement $1 + \kappa_{cos}$ rather than $\kappa_{cos}$.

**Definition 30** (Gaussian kernel)**.** *Given a data set $V \in \mathbb{R}^m$ for some $m \geq 0$, the Gaussian kernel $\kappa_{gauss}$ with standard deviation $\sigma$ is defined as $\kappa_{gauss}(x, y) \coloneqq \exp\left(-\frac{\|x-y\|^2}{2\sigma^2}\right)$.*

The main aim of our experiments was to answer the following two questions.

1. How good is the hierarchal clustering obtained from Algorithm 4 as opposed to the true optimal output by f-ILP-ultrametric?

2. How good does Algorithm 4 perform compared to other hierarchical clustering methods?

For the first question, we are restricted to working with small data sets since computing an optimum solution to f-ILP-ultrametric is expensive. In this case we consider synthetic data sets of small size and samples of some data sets from the UCI database [36]. The synthetic data sets we consider are mixtures of Gaussians in various small dimensional spaces. Figure 1 shows a comparison of the cost of the hierarchy (according to cost function (26)) returned by solving f-ILP-ultrametric and by Algorithm 4 for various forms of $f$ when the similarity function is $\kappa_{cos}$ and $\kappa_{gauss}$. Note that we normalize the cost of the tree returned by f-ILP-ultrametric and Algorithm 4 by the cost of the trivial clustering $r, T^*$ where $T^*$ is the star graph with $V$ as its leaves and $r$ as the internal node. In other words $d_{T^*}(i, j) = n - 1$ for every distinct pair $i, j \in V$ and so the normalized cost of any tree lies in the interval $(0, 1]$.

For the study of the second question, we consider some of the popular algorithms for hierarchical clustering are *single linkage*, *average linkage*, *complete linkage*, and *Ward's method* [45]. To get a numerical handle on how good a hierarchical clustering $T$ of $V$ is, we prune the tree to get the *best $k$ flat clusters* and measure its error relative to the target clustering. We use the following notion of error also known as *Classification Error* that is standard in the literature for hierarchical clustering (see, e.g., [37]). Note that we may think of a flat $k$-clustering of the data $V$ as a function $h$ mapping elements of $V$ to a label set $\mathcal{L} \coloneqq \{1, \ldots, k\}$. Let $S_k$ denote the group of permutations on $k$ letters.

**Definition 31** (Classification Error)**.** *Given a proposed clustering $h : V \to \mathcal{L}$ its* classification error *relative to a target clustering $g : V \to \mathcal{L}$ is denoted by $\mathrm{err}\,(g, h)$ and is defined as*

$$\mathrm{err}\,(g, h) \coloneqq \min_{\sigma \in S_k}\left[\Pr_{x \in V}[h(x) \neq \sigma(g(x))]\right].$$

**Example 32.** *Recall the data set from Example **??**. Let $k = 3$ and $g$ be the target clustering defined as $g(x_0) = g(x_1) = 1$, $g(x_2) = g(x_3) = 2$, and $g(x_4) = g(x_5) = 3$. Then the error of the best pruning of the hierarchal clustering in Figures **??** and **??** is $0$ while for Figures **??** and **??** it is $\frac{1}{6}$.*

We compare the error of Algorithm 4 with the various linkage based algorithms that are commonly used for hierarchical clustering, as well as Ward's method and the $k$-means algorithm. We test Algorithm 4 most extensively for $f(x) = x$ while doing a smaller number of tests for $f(x) \in \left\{x^2, \log(1+x), e^x - 1\right\}$. Note that both Ward's method and the $k$-means algorithm work on the squared Euclidean distance $\|x - y\|_2^2$ between two points $x, y \in V$, i.e., they both require an embedding of the data points into a normed vector space which provides extra information that

Figure 1: Comparison of f-ILP-ultrametric and Algorithm 4 for $1 + \kappa_{cos}$ (left) and $\kappa_{gauss}$ (right)

Figure 2: Comparison of Algorithm 4 using $f(x) = x$, with other algorithms for clustering using $1 + \kappa_{cos}$ (left) and $\kappa_{gauss}$ (right)

can be potentially exploited. For the linkage based algorithms we use the same notion of similarity $1 + \kappa_{cos}$ or $\kappa_{gauss}$ that we use for Algorithm 4. For comparison we use a mix of synthetic data sets as well as the Wine, Iris, Soybean-small, Digits, Glass, and Wdbc data sets from the UCI repository [36]. For some of the larger data sets, we sample uniformly at random a smaller number of data points and take the average of the error over the different runs. Figures 2, 3, 4, and 5 show that the hierarchical clustering returned by Algorithm 4 with $f(x) \in \{x, x^2, \log(1 + x), e^x - 1\}$ often has better projections into flat clusterings than the other algorithms. This is especially true when we compare it to the linkage based algorithms, since they use the same pairwise similarity function as Algorithm 4, as opposed to Ward's method and $k$-means.

## 7 Discussion

In this work we have studied the cost functions (1) and (26) for hierarchical clustering given a pairwise similarity function over the data and shown an $O(\log n)$ approximation algorithm for this problem. As briefly mentioned in Section 2 however, such a cost function is not unique. Further, there is an intimate connection between hierarchical clusterings and ultrametrics over discrete sets which points to other directions for formulating a cost function over hierarchies. In particular we briefly mention the related notion of *hierarchically well-separated trees* (HST) as defined in [6] (see also [8, 9]). A $k$-HST for $k \geq 1$ is a tree $T$ such that each vertex $u \in T$ has a label $\Delta(u) \geq 0$ such

Figure 3: Comparison of Algorithm 4 using $f(x) = x^2$, with other algorithms for clustering using $1 + \kappa_{cos}$ (left) and $\kappa_{gauss}$ (right)

Figure 4: Comparison of Algorithm 4 using $f(x) = \log(1+x)$, with other algorithms for clustering using $1 + \kappa_{cos}$ (left) and $\kappa_{gauss}$ (right)

Figure 5: Comparison of Algorithm 4 using $f(x) = e^x - 1$, with other algorithms for clustering using $1 + \kappa_{cos}$ (left) and $\kappa_{gauss}$ (right)

that $\Delta(u) = 0$ if and only if $u$ is a leaf of $T$. Further, if $u$ is a child of $v$ in $T$ then $\Delta(u) \leq \Delta(v)/k$. It is well known that any ultrametric $d$ on a finite set $V$ is equivalent to a 1-HST where $V$ is the set of leaves of $T$ and $d(i,j) = \Delta(\text{lca}(i,j))$ for every $i, j \in V$. Thus in the special case when $\Delta(u) = |\text{leaves } T[u]| - 1$ we get the cost function (1), while if $\Delta(u) = f(|\text{leaves } T[u]| - 1)$ for a strictly increasing function $f$ with $f(0) = 0$ then we get cost function (26). It turns out this assumption on $\Delta$ enables us to prove the combinatorial results of Section 3 and give a $O(\log n)$ approximation algorithm to find the optimal cost tree according to these cost functions. It is an interesting problem to investigate cost functions and algorithms for hierarchical clustering induced by other families of $\Delta$ that arise from a $k$-HST on $V$, i.e., if the cost of $T$ is defined as

$$\text{cost}_\Delta(T) := \sum_{\{i,j\} \in E(K_n)} \kappa(i,j) \Delta(\text{lca}(i,j)). \tag{45}$$

Note that not all choices of $\Delta$ lead to a meaningful cost function. For example, choosing $\Delta(u) = \text{diam}(T[u]) - 1$ gives rise to the following cost function

$$\text{cost}(T) := \sum_{\{i,j\} \in E(K_n)} \kappa(i,j) \text{dist}_T(i,j) \tag{46}$$

where $\text{dist}_T(i,j)$ is the length of the unique path from $i$ to $j$ in $T$. In this case, the trivial clustering $r, T^*$ where $T^*$ is the star graph with $V$ as its leaves and $r$ as the root is always a minimizer; in other words, there is no incentive for spreading out the hierarchical clustering. Also worth mentioning is a long line of related work on fitting tree metrics to metric spaces (see e.g., [2, 40, 21]). In this setting, the data points $V$ are assumed to come from a metric space $d_V$ and the objective is to find a hierarchical clustering $T$ so as to minimize $\|d_V - d_T\|_p$. If the points in $V$ lie on the unit sphere and the similarity function $\kappa$ is the cosine similarity $\kappa_{cos}(i,j) = 1 - d_V(i,j)/2$, then the problem of fitting a tree metric with $p = 2$ minimizes the same objective as cost function (46). Since $d_V \leq 1$ in this case, the minimizer is the trivial tree $r, T^*$ (as remarked above). In general, when the points in $V$ are not constrained to lie on the unit sphere, the two problems are incomparable.

## 8 Acknowledgments

Research reported in this paper was partially supported by NSF CAREER award CMMI-1452463 and NSF grant CMMI-1333789. We would like to thank Kunal Talwar and Mohit Singh for helpful discussions and anonymous reviewers for helping improve the presentation of this paper.

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

## A   Hardness of finding the optimal hierarchical clustering

In this section we study the hardness of finding the optimal hierarchical clustering according to cost function (1). We show that under the assumption of the *Small Set Expansion* (SSE) hypothesis there is no constant factor approximation algorithm for this problem. We also show that no polynomial sized Linear Program (LP) or Semidefinite Program (SDP) can give a constant factor approximation for this problem without the need for any complexity theoretic assumptions. Both these results make use of the similarity of this problem with the *minimum linear arrangement* problem. To show hardness under Small Set Expansion, we make use of the result of [41] showing that there is no constant factor approximation algorithm for the Minimum Linear Arrangement problem under

the assumption of SSE. To show the LP and SDP inapproximability results, we make use of the reduction framework of [10] together with the NP-hardness proof for Minimum Linear Arrangement due to [24]. We also note that both these hardness results hold even for unweighted graphs (i.e., when $\kappa \in \{0, 1\}$).

Note that the individual layer-$t$ problem f-ILP-layer for $t = \lfloor n/2 \rfloor$ is equivalent to the *minimum bisection problem* for which the best known approximation is $O(\log n)$ due to [40], while the best known bi-criteria approximation is $O\left(\sqrt{\log n}\right)$ due to [3] and improving these approximation factors is a major open problem. However it is not clear if an improved approximation algorithm for hierarchical clustering under cost function (1) would imply an improved algorithm for every layer-$t$ problem, which is why a constant factor inapproximability result is of interest. We start by recalling the definition of an *optimization problem* in the framework of [10].

**Definition 33** (Optimization problem). *[10] An* optimization problem *is a tuple* $\mathcal{P} = (\mathcal{S}, \mathfrak{I}, \mathrm{val})$ *consisting of a set* $\mathcal{S}$ *of* feasible solutions*, a set* $\mathfrak{I}$ *of* instances*, and a real-valued objective called* measure $\mathrm{val} \colon \mathfrak{I} \times \mathcal{S} \to \mathbb{R}$*. We shall use* $\mathrm{val}_{\mathcal{I}}(s)$ *for the objective value of a feasible solution* $s \in \mathcal{S}$ *for an instance* $\mathcal{I} \in \mathfrak{I}$*.*

Since we are interested in the integrality gaps of LP and SDP relaxations for an optimization problem $\mathcal{P} = (\mathcal{S}, \mathfrak{I}, \mathrm{val})$, we represent the approximation gap by two functions $C, S : \mathfrak{I} \to \mathbb{R}$ where $C$ is the *completeness guarantee* while $S$ is the *soundness guarantee*. Note that the ratio $C/S$ represents the approximation factor for the problem $\mathcal{P}$. We recall below the formal definition of an LP relaxation of $\mathcal{P}$ that achieves a $(C, S)$-approximation guarantee. We assume without loss of generality that $\mathcal{P}$ is a maximization problem.

**Definition 34** (LP formulation of an optimization problem). *[10] Let* $\mathcal{P} = (\mathcal{S}, \mathfrak{I}, \mathrm{val})$ *be an optimization problem, and* $C, S : \mathfrak{I} \to \mathbb{R}$*. Then let* $\mathfrak{I}^S := \{\mathcal{I} \in \mathfrak{I} \mid \max \mathrm{val}_{\mathcal{I}} \leq S(\mathcal{I})\}$ *denote the set of* sound *instances, i.e., for which the soundness guarantee* $S$ *is an upper bound on the maximum. A* $(C, S)$-approximate LP formulation *of* $\mathcal{P}$ *consists of a linear program* $Ax \leq b$ *with* $x \in \mathbb{R}^r$ *for some* $r$ *and the following* realizations*:*

**Feasible solutions** *as vectors* $x^s \in \mathbb{R}^r$ *for every* $s \in \mathcal{S}$ *satisfying*

$$Ax^s \leq b \qquad \text{for all } s \in \mathcal{S}, \tag{47}$$

*i.e., the system* $Ax \leq b$ *is a relaxation of* $\mathrm{conv}\,(x^s \mid s \in \mathcal{S})$.

**Instances** *as affine functions* $w_{\mathcal{I}} \colon \mathbb{R}^r \to \mathbb{R}$ *for all* $\mathcal{I} \in \mathfrak{I}^S$ *satisfying*

$$w_{\mathcal{I}}(x^s) = \mathrm{val}_{\mathcal{I}}(s) \qquad \text{for all } s \in \mathcal{S}, \tag{48}$$

*i.e., the linearization* $w_{\mathcal{I}}$ *of* $\mathrm{val}_{\mathcal{I}}$ *is required to be exact on all* $x^s$ *with* $s \in \mathcal{S}$*.*

**Achieving** $(C, S)$ **approximation guarantee** *by requiring*

$$\max \{w_{\mathcal{I}}(x) \mid Ax \leq b\} \leq C(\mathcal{I}) \qquad \text{for all } \mathcal{I} \in \mathfrak{I}^S, \tag{49}$$

*The* size *of the formulation is the number of inequalities in* $Ax \leq b$*. Finally, the* $(C, S)$-approximate LP formulation complexity $\mathrm{fc}_{\mathrm{LP}}(\mathcal{P}, C, S)$ *of* $\mathcal{P}$ *is the minimal size of all its LP formulations.*

One can similarly define a $(C, S)$-approximate SDP formulation for a problem $\mathcal{P}$ where instead of a LP, we now have a SDP relaxation $\mathcal{A}(X) = b$ with $X \in \mathbb{S}^r_+$ and where $\mathbb{S}^r_+$ denotes the space of $r \times r$ positive semidefinite matrices. The size of such an SDP formulation is measured by the dimension $r$ and $\mathrm{fc}_{\mathrm{SDP}}(\mathcal{P}, C, S)$ is defined as the minimum size of an SDP formulation achieving $(C, S)$-approximation for problem $\mathcal{P}$. Below we recall the precise notion of a reduction between two problems as in [10].

**Definition 35** (Reduction). *[10] Let* $\mathcal{P}_1 = (\mathcal{S}_1, \mathfrak{I}_1, \mathrm{val})$ *and* $\mathcal{P}_2 = (\mathcal{S}_2, \mathfrak{I}_2, \mathrm{val})$ *be optimization problems with guarantees* $C_1, S_1$ *and* $C_2, S_2$*, respectively. Let* $\tau_1 = +1$ *if* $\mathcal{P}_1$ *is a maximization problem, and* $\tau_1 = -1$ *if* $\mathcal{P}_1$ *is a minimization problem. Similarly, let* $\tau_2 = \pm 1$ *depending on whether* $\mathcal{P}_2$ *is a maximization problem or a minimization problem.*

*A* reduction *from* $\mathcal{P}_1$ *to* $\mathcal{P}_2$ *respecting the guarantees consists of*

1. *two mappings:* $* \colon \mathfrak{I}_1 \to \mathfrak{I}_2$ *and* $* \colon \mathcal{S}_1 \to \mathcal{S}_2$ *translating instances and feasible solutions independently;*

2. *two nonnegative $\mathfrak{I}_1 \times \mathcal{S}_1$ matrices $M_1$, $M_2$*

*subject to the conditions*

$$\tau_1\left[C_1(\mathcal{I}_1) - \mathrm{val}_{\mathcal{I}_1}(s_1)\right] = \tau_2\left[C_2(\mathcal{I}_1^*) - \mathrm{val}_{\mathcal{I}_1^*}(s_1^*)\right]M_1(\mathcal{I}_1, s_1) + M_2(\mathcal{I}_1, s_1) \quad \text{(50-complete)}$$

$$\tau_2\,\mathrm{OPT}\,(\mathcal{I}_1^*) \le \tau_2 S_2(\mathcal{I}_1^*) \qquad \textit{if } \tau_1\,\mathrm{OPT}\,(\mathcal{I}_1) \le \tau_1 S_1(\mathcal{I}_1). \qquad\qquad \text{(50-sound)}$$

The matrices $M_1$ and $M_2$ control the parameters of the reduction relating the integrality gap of relaxations for $\mathcal{P}_1$ to the integrality gap of corresponding relaxations for $\mathcal{P}_2$. For a matrix $A$, let $\mathrm{rk}_+\,A$ and $\mathrm{rk}_{\mathrm{psd}}\,A$ denote the nonnegative rank and psd rank of $A$ respectively. The following theorem is a restatement of Theorem 3.2 from [10] ignoring constants.

**Theorem 36.** *[10] Let $\mathcal{P}_1$ and $\mathcal{P}_2$ be optimization problems with a reduction from $\mathcal{P}_1$ to $\mathcal{P}_2$ respecting the completeness guarantees $C_1$, $C_2$ and soundness guarantees $S_1$, $S_2$ of $\mathcal{P}_1$ and $\mathcal{P}_2$, respectively. Then*

$$\mathrm{fc}_{\mathrm{LP}}(\mathcal{P}_1, C_1, S_1) \le \mathrm{rk}_+\,M_2 + \mathrm{rk}_+\,M_1 + \mathrm{rk}_+\,M_1 \cdot \mathrm{fc}_{\mathrm{LP}}(\mathcal{P}_2, C_2, S_2), \qquad (51)$$

$$\mathrm{fc}_{\mathrm{SDP}}(\mathcal{P}_1, C_1, S_1) \le \mathrm{rk}_{\mathrm{psd}}\,M_2 + \mathrm{rk}_{\mathrm{psd}}\,M_1 + \mathrm{rk}_{\mathrm{psd}}\,M_1 \cdot \mathrm{fc}_{\mathrm{SDP}}(\mathcal{P}_2, C_2, S_2), \qquad (52)$$

*where $M_1$ and $M_2$ are the matrices in the reduction as in Definition 35.*

Therefore to obtain a lower bound for problem $\mathcal{P}_2$, it suffices to find a source problem $\mathcal{P}_1$ and matrices $M_1$ and $M_2$ of low nonnegative rank and low psd rank, satisfying Definition 35.

Below, we cast the hierarchical clustering problem (HCLUST) as an optimization problem. We also recall a different formulation of cost function (1) due to [16] that will be useful in the analysis of the reduction.

**Definition 37** (HCLUST as optimization problem). *The minimization problem HCLUST of size $n$ consists of*

**instances** *similarity function $\kappa : E(K_n) \to \mathbb{R}_{\ge 0}$*

**feasible solutions** *hierarchical clustering $r, T$ of $V(K_n)$*

**measure** $\mathrm{val}_\kappa(T) = \sum_{\{i,j\}\in E(K_n)} \kappa(i,j)\,|\mathrm{leaves}(T[\mathrm{lca}(i,j)])|.$

We will also make use of the following alternate interpretation of cost function (1) given by [16]. Let $\kappa : V \times V \to \mathbb{R}_{\ge 0}$ be an instance of HCLUST. For a subset $S \subseteq V$, a split $S_1, \ldots, S_k$ is a partition of $S$ into $k$ disjoint pieces. For a binary split $S_1, S_2$ we can define $\kappa(S_1, S_2) := \sum_{i\in S_1, j\in S_2}\kappa(i,j)$. This can be extended to $k$-way splits in the natural way:

$$\kappa(S_1, \ldots, S_k) := \sum_{1\le i\le j\le k}\kappa(S_i, S_j).$$

Then the cost of a tree $T$ is the sum over all the internal nodes of the splitting costs at the nodes, as follows.

$$\mathrm{cost}(T) = \sum_{\text{splits } S \to (S_1,\ldots,S_k) \text{ in } T} |S|\,\kappa(S_1, \ldots, S_k).$$

We now briefly recall the MAXCUT problem.

**Definition 38** (MAXCUT as optimization problem). *The maximization problem MAXCUT of size $n$ consists of*

**instances** *all graphs $G$ with $V(G) \subseteq [n]$*

**feasible solutions** *all subsets $X$ of $[n]$*

**measure** $\mathrm{val}_G(X) = |\delta_G(X)|.$

Similarly, the Minimum Linear Arrangement problem can be phrased as an optimization problem as follows.

**Definition 39** (MLA as optimization problem). *The minimization problem* MLA *of size $n$ consists of*

**instances** *weight function $w : E(K_n) \to \mathbb{R}_{\geq 0}$*

**feasible solutions** *all permutations $\pi : V(K_n) \to [n]$*

**measure** $\mathrm{val}_w(\pi) := \sum_{\{i,j\} \in E(K_n)} w(i,j)\, |\pi(i) - \pi(j)|$.

We now describe the reduction from MAXCUT to HCLUST which is a modification of the reduction from MAXCUT to MLA due to [24]. Note that an instance of MAXCUT maps to an unweighted instance of HCLUST, i.e., $\kappa \in \{0,1\}$.

**Mapping instances** Given an instance $G = (V, E)$ of MAXCUT of size $n$, let $r = n^4$ and $U = \{u_1, u_2, \ldots, u_r\}$. The instance $\kappa$ of HCLUST is on the graph with vertex set $V' := V \cup U$ and has weights in $\{0, 1\}$. For any distinct pair $i, j \in V'$, if $\{i, j\} \in E$ then we define $\kappa(i, j) := 0$ and otherwise we set $\kappa(i, j) := 1$.

**Mapping solutions** Given a cut $X \subseteq V$ of MAXCUT we map it to the clustering $r, T$ of $V'$ where the root $r$ has the following children: $n^4$ leaves corresponding to $U$, and 2 internal vertices corresponding to $X$ and $\overline{X}$. The internal vertices for $X$ and $\overline{X}$ are split into $|X|$ and $\left|\overline{X}\right|$ leaves respectively at the next level.

The following lemma relates the LP and SDP formulations for MAXCUT and MLA.

**Lemma 40.** *For any completeness and soundness guarantee $(C, S)$, we have the following*

$$\mathrm{fc}_{\mathrm{LP}}\left(\mathsf{MAXCUT}, C, S\right) \leq \mathrm{fc}_{\mathrm{LP}}\left(\mathsf{HCLUST}, C', S'\right) + O(n^2)$$

$$\mathrm{fc}_{\mathrm{SDP}}\left(\mathsf{MAXCUT}, C, S\right) \leq \mathrm{fc}_{\mathrm{SDP}}\left(\mathsf{HCLUST}, C', S'\right) + O(n^2).$$

*where $C' := \frac{(n^4+n)^3 - (n^4+n)}{3} - C(n^4 + n)$ and $S' := \binom{n^4+n+1}{3} - Sn^4$.*

*Proof.* To show completeness, we analyze the cost of the tree $T$ that a cut $X$ maps to, using the alternate interpretation of the cost function (1) due to [16] (see above). Let $H$ be the graph on vertex set $V'$ induced by $\kappa$, i.e. $\{i, j\} \in E(H)$ iff $\kappa(i, j) = 1$. Let $\overline{H}$ denote the complement graph of $H$ and let $\overline{\kappa}$ be the similarity function induced by it, i.e., $\overline{\kappa}(i, j) = 1$ iff $\{i, j\} \notin E(H)$ and $\overline{\kappa}(i, j) = 0$ otherwise. For a hierarchical clustering $T$ of $V'$, we denote by $\mathrm{cost}_H(T)$ and $\mathrm{cost}_{\overline{H}}(T)$ the cost of $T$ induced by $\kappa$ and $\overline{\kappa}$ respectively, i.e., $\mathrm{cost}_H(T) := \sum_{\{i,j\} \in E(H)} |\mathrm{leaves}(T[\mathrm{lca}(i,j)])|$ and $\mathrm{cost}_{\overline{H}}(T) := \sum_{\{i,j\} \notin E(H)} |\mathrm{leaves}(T[\mathrm{lca}(i,j)])|$. Let $\overline{X} := V' \setminus X$. The cost of the tree $T$ that the cut $X$ maps to, is given by

$$
\begin{aligned}
\mathrm{cost}(T) &= \mathrm{cost}_H(T) \\
&= \frac{\left(n + n^4\right)^3 - \left(n + n^4\right)}{3} - \mathrm{cost}_{\overline{H}}(T) \\
&= \frac{\left(n + n^4\right)^3 - \left(n + n^4\right)}{3} - \sum_{\text{splits } S \to (S_1, \ldots, S_k) \text{ in } T} |S|\, \overline{\kappa}(S_1, \ldots, S_k) \\
&= \frac{\left(n + n^4\right)^3 - \left(n + n^4\right)}{3} - \left(n + n^4\right) \mathrm{val}_G(X) - \left(|X|\,|E[X]| + \left|\overline{X}\right|\,\left|E[\overline{X}]\right|\right),
\end{aligned}
$$

where $E[X]$ and $E[\overline{X}]$ are the edges of $E(H)$ induced on the set $X$ and $\overline{X}$ respectively. Therefore, we have the following completeness relationship between the two problems

$$C - \mathrm{val}_G(X) = \frac{1}{n + n^4}\left(\mathrm{cost}(T) - \left(\frac{(n+n^4)^3 - (n+n^4)}{3} - C(n + n^4)\right)\right) + \frac{|X|\,|E[X]| + \left|\overline{X}\right|\,\left|E[\overline{X}]\right|}{n^4 + n}.$$

We now define the matrices $M_1$ and $M_2$ as $M_1(H, X) := \frac{1}{n+n^4}$ and $M_2(H, X) := |X|\,|E[X]| + \left|\overline{X}\right|\,\left|E[\overline{X}]\right|$. Clearly, $M_1$ has $O(1)$ nonnegative rank and psd rank. We claim that the nonnegative

rank of $M_2$ is at most $2\binom{n}{2}$. The vectors $v_H \in \mathbb{R}^{2\binom{n}{2}}$ corresponding to the instances $H$ is defined as the concatenation $[u_H, w_H]$ of two vectors $u_H, w_H \in \mathbb{R}^{\binom{n}{2}}$. Both the vectors $u_H, w_H$ encode the edges of $H$ scaled by $n^4 + n$, i.e., $u_H(\{i,j\}) = w_H(\{i,j\}) = 1/(n^4 + n)$ iff $\{i,j\} \in E(H)$ and 0 otherwise. The vectors $v_X \in \mathbb{R}^{2\binom{n}{2}}$ corresponding to the solutions are also defined as the concatenation $[u_X, w_X]$ of two vectors $u_X, w_X \in \mathbb{R}^n$. The vector $u_X$ encodes the vertices in $X$ scaled by $|X|$ i.e., $u_X(\{i,j\}) = |X|$ iff $i, j \in X$ and 0 otherwise. The vector $w_X$ encodes the vertices in $\overline{X}$ scaled by $|\overline{X}|$ i.e., $w_X(\{i,j\}) = |\overline{X}|$ iff $i, j \in \overline{X}$ and 0 otherwise. Clearly, we have $M_2(H, X) = \langle v_H, v_X \rangle$ and so the nonnegative (and psd) rank of $M_2$ is at most $2\binom{n}{2}$.

Soundness follows due to the analysis in [24] and by noting that the cost of a linear arrangement obtained by projecting the leaves of $T$ is a lower bound on $\text{cost}(T)$. By the analysis in [24] if the optimal value $\text{OPT}(G)$ of MAXCUT is at most $S$, then the optimal value of MLA on $V', \kappa$ is at least $\binom{n^4 + n + 1}{3} - Sn^4$. Therefore, it follows that the optimal value of HCLUST on $V', \kappa$ is also at least $\binom{n^4 + n + 1}{3} - Sn^4$. $\qquad\square$

The constant factor inapproximability result for HCLUST now follows due to the following theorems.

**Theorem 41** ([11, Theorem 3.2]). *For any $\varepsilon > 0$ there are infinitely many $n$ such that*

$$\text{fc}_{\text{LP}}\left(\text{MAXCUT}, 1 - \varepsilon, \frac{1}{2} + \frac{\varepsilon}{6}\right) \geq n^{\Omega(\log n / \log \log n)}.$$

**Theorem 42** ([10, Theorem 7.1]). *For any $\delta, \varepsilon > 0$ there are infinitely many $n$ such that*

$$\text{fc}_{\text{SDP}}\left(\text{MAXCUT}, \frac{4}{5} - \varepsilon, \frac{3}{4} + \delta\right) = n^{\Omega(\log n / \log \log n)}. \tag{53}$$

Thus we have the following corollary about the LP and SDP inapproximability for the problem HCLUST.

**Corollary 43** (LP and SDP hardness for HCLUST). *For any constant $c \geq 1$, HCLUST is LP-hard and SDP-hard with an inapproximability factor of $c$.*

*Proof.* Straightforward by using Theorems 41 and 42 together with Lemma 40 and by choosing $n$ large enough. $\qquad\square$

The following lemma shows that a minor modification of the argument in [41] also implies a constant factor inapproximability result under the *Small Set Expansion* (SSE) hypothesis. Note that this reduction is also true for unit capacity graphs, i.e., $\kappa \in \{0, 1\}$. We briefly recall the formulation of the Small Set Expansion hypothesis. Informally, given a graph $G = (V, E)$ the problem is to decide whether all "small" sets in the graph are expanding. Let $d(i)$ denote the degree of a vertex $i \in V$. For a subset $S \subseteq V$ let $\mu(S) := |S| / |V|$ be the volume of $S$, and let $\phi(S) := E(S, \overline{S}) / \sum_{i \in S} d(i)$ be the expansion of $S$. Then the SSE problem is defined as follows.

**Definition 44** (Small set expansion (SSE) hypothesis [41]). *For every constant $\eta > 0$, there exists sufficiently small $\delta > 0$ such that given a graph $G = (V, E)$, it is NP-hard to decide the following cases,*

**Completeness** *there exists a subset $S \subseteq V$ with volume $\mu(S) = \delta$ and expansion $\phi(S) \leq \eta$,*

**Soundness** *every subset $S \subseteq V$ of volume $\mu(S) = \delta$ has expansion $\phi(S) \geq 1 - \eta$.*

Under this assumption, [41] proved the following amplification result about the expansion of small sets in the graph.

**Theorem 45** (Theorem 3.5 [41]). *For all $q \in \mathbb{N}$ and $\varepsilon', \gamma > 0$ it is SSE-hard to distinguish the following for a given graph $H = (V_H, E_H)$*

**Completeness** *There exist disjoint sets $S_1, \ldots, S_q \subseteq V_H$ satisfying $\mu(S_i) = \frac{1}{q}$ and $\phi(S_i) \leq \varepsilon' + o(\varepsilon')$ for all $i \in [n]$,*

**Soundness** *For all sets $S \subseteq V_H$ we have $\phi(S) \geq \phi_{\mathcal{G}}(1 - \varepsilon'/2)(\mu(S)) - \gamma/\mu(S)$,*

*where $\phi_{\mathcal{G}}(1 - \varepsilon'/2)(\mu(S))$ is the expansion of sets of volume $\mu(S)$ in the infinite Gaussian graph $\mathcal{G}(1 - \varepsilon'/2)$.*

The following lemma establishes that it is SSE-hard to approximate HCLUST to within any constant factor. The argument closely parallels Corollary A.5 of [41] where it was shown that it is SSE-hard to approximate MLA to within any constant factor.

**Lemma 46.** *Let $G = (V, E)$ be a graph on $V$ with $\kappa$ induced by the edges $E$ i.e., $\kappa(i,j) = 1$ iff $\{i,j\} \in E$ and $0$ otherwise. Then it is SSE-hard to distinguish between the following two cases*

**Completeness** *There exists a hierarchical clustering $T$ of $V$ with $\mathrm{cost}(T) \leq \varepsilon n \, |E|$,*

**Soundness** *Every hierarchical clustering $T$ of $V$ satisfies $\mathrm{cost}(T) \geq c\sqrt{\varepsilon} n \, |E|$*

*for some constant $c$ not depending on $n$.*

*Proof.* Apply Theorem 45 on the graph $G$ with the following choice of parameters: $q = \lceil 2/\varepsilon \rceil$, $\varepsilon' = \varepsilon/3$ and $\gamma = \varepsilon$. Suppose there exist $S_1, \ldots, S_q \subseteq V$ satisfying $\phi(S_i) \leq \varepsilon' + o(\varepsilon')$ and $|S_i| = |V|/q \leq \varepsilon \, |V|/2$. Then consider the tree $r, T$ with the root $r$ having $q$ children corresponding to each $S_i$, and each $S_i$ being further separated into $|S_i|$ leaves at the next level. We claim that $\mathrm{cost}(T) \leq \varepsilon n \, |E|$. We analyze this using the alternate interpretation of cost function (1) (see above). Every crossing edge between $S_i, S_j$ for distinct $i, j \in [q]$ incurs a cost of $n$, but by assumption there are at most $\varepsilon \, |E|/2$ such edges. Further, any edge in $S_i$ incurs a cost $\frac{n}{q} \leq \varepsilon n/2$ and thus their contribution is upper bounded by $\varepsilon n \, |E|$.

The analysis for soundness follows by the argument of Corollary A.5 in [41]. In particular, if for every $S \subseteq V$ we have $\phi(S) \geq \phi_{\mathcal{G}}(1 - \varepsilon'/2)(\mu(S)) - \gamma/\mu(S)$ then the cost of the optimal linear arrangement on $G$ is at most $\sqrt{\varepsilon} n \, |E|$. Since the cost of any tree (including the optimal tree) is at least the cost of the linear arrangement induced by projecting the leaf vertices, the claim about soundness follows. $\qquad\square$