[Reviews · NeurIPS 2016]

Reviewer 1

Summary

The paper presents an improved algorithm for the recent Hierarchical Clustering problem that was introduced by Dasgupta [STOC 16]: there are n vertices, with a similarity weight between every pair of vertices. The goal is to find a hierarchical clustering (tree) the minimizes a well-defined objective. This gives a nice objective to compare and design algorithms for hierarchical clustering. The paper gives an algorithm that always finds solutions of cost O(log n).OPT i.e., a O(log n) approximation, that improves upon the result of Dasgupta that gave a O(\log^{3/2} n) factor approximation. They also give experimental results demonstrating the advantage of their approach for small input sizes, and also give evidence that no LP-based approach can give O(1) factor approximations. The techniques used in this paper are nice and insightful: the authors characterize these hierarchical clustering in terms of ultrametrics with some nice, new properties of ultrametrics, and write down an LP relaxation to find such a ultrametric. Using somewhat standard techniques in approximate graph partitioning, they obtain the O(log n) approximation.

Qualitative Assessment

The paper of Dasgupta introduced a nice objective for hierarchical clustering, and presented a fairly simple algorithm for approximating the objective. I like this objective since it gives a nice principled way of comparing different algorithms (potentially new ones) for hierarchical clustering, as opposed to just showing bounds for linkage-based heuristics. I think the algorithm and analysis in this paper is very elegant. Every hierarchical clustering corresponds to an ultrametric; but they characterize the ultrametrics that correspond to the objective defined by Dasgupta, using some very nice properties. They use this characterization to come up with an LP and its rounding. A couple of drawbacks: while this algorithm seems to perform better experimentally, it is ineffective for large instances because of the size of the LP. Secondly, the rounding is fairly standard in approximation algorithms. However, I believe that the paper has a very nice set of new ideas in the context of hierarchical clustering, and hence should be accepted. Comments: 1. A paper by Ailon and Charikar (on fitting tree metrics) uses a somewhat similar LP (but seemingly weaker LP) with variables for different levels, and uses a similar rounding approach. It would be good to reference and compare this result. 2. In page 4, line 158: "To see this, note that the constraint is active only when .... ". This seems to need more of an explanation to say that \sum_{i,j \in S} x^t_{ij} <= |S| always.

Confidence in this Review

2-Confident (read it all; understood it all reasonably well)


Reviewer 2

Summary

I enjoyed reading this paper which shows that the hierarchical clustering problem [6] can be approximated within a multiplicative O(log n) factor. This improves the O(\alpha log n) approximation of [6] where \alpha is an approximation factor of a subroutine used to solve the “Sparsest Cut” problem. In order to avoid losing the approximation factor of the “Sparsest Cut” problem the authors develop a new approach that is based on a careful LP rounding algorithm. In a nutshell, an ILP formulation is used to give a characterization of ultrametrics associated with the problem. The key observation here is that the ultrametrics associated with the hierarchical clustering problem have to satisfy a non-triviality condition (Def 1 + Lem 5). This condition is then captured by the ILP together with the ultrametric condition (for the latter the techniques of [8] are used). Then an LP-relaxation based algorithm with iterative rounding is given that achieves the desired approximation using Leighton-Rao “sphere growing” technique. It is quite remarkable that a fairly complex LP-formulation used in this work does in fact admit such a rounding approach. I also very much enjoyed the overall comprehensiveness of the approach taken in this paper as it includes experimental evaluation on datasets from the UCI database + negative results that show superpolynomial lower bounds on the LP/SDP relaxations of the problem.

Qualitative Assessment

I enjoyed reading this paper which shows that the hierarchical clustering problem [6] can be approximated within a multiplicative O(log n) factor. This improves the O(\alpha log n) approximation of [6] where \alpha is an approximation factor of a subroutine used to solve the “Sparsest Cut” problem. In order to avoid losing the approximation factor of the “Sparsest Cut” problem the authors develop a new approach that is based on a careful LP rounding algorithm. In a nutshell, an ILP formulation is used to give a characterization of ultrametrics associated with the problem. The key observation here is that the ultrametrics associated with the hierarchical clustering problem have to satisfy a non-triviality condition (Def 1 + Lem 5). This condition is then captured by the ILP together with the ultrametric condition (for the latter the techniques of [8] are used). Then an LP-relaxation based algorithm with iterative rounding is given that achieves the desired approximation using Leighton-Rao “sphere growing” technique. It is quite remarkable that a fairly complex LP-formulation used in this work does in fact admit such a rounding approach. I also very much enjoyed the overall comprehensiveness of the approach taken in this paper as it includes experimental evaluation on datasets from the UCI database + negative results that show superpolynomial lower bounds on the LP/SDP relaxations of the problem.

Confidence in this Review

3-Expert (read the paper in detail, know the area, quite certain of my opinion)


Reviewer 3

Summary

The paper studies the Hierarchical Clustering problem with the objective introduced by Dasgupta [6]. It gives a new O(log n)-approximation algorithm for the problem. The original algorithm by Dasgupta gives O(log n)^3/2 approximation. The new algorithm casts the hierarchical clustering problem as a problem of finding an ultrametric satisfying spreading constraints. It solves a linear program to find a fractional solution, and then rounds it using the standard sphere growing technique.

Qualitative Assessment

The technical part of the paper is fairly standard. It is a borderline submission for NIPs.

Confidence in this Review

2-Confident (read it all; understood it all reasonably well)


Reviewer 4

Summary

This paper is looking for a faster algorithm by doing layer based LP relaxation for an Integer Linear Programming(ILP) formulation. As I see, the authors provided theoretical proofs for this better complexity of their algorithm, and shown the performance experimentally.

Qualitative Assessment

As I known Hierarchical clustering is always one of the most popular clustering methods so far, but few literature focuses on theoretical analysis on it, the story of this paper is complete, but there is still advancing room, for example, it seems that it is not true that the smaller the cost function is the better as show in experiment. And I have a suggestion, can you show the running time of your algorithm experimentally, since one of your theoretical contribution is the complexity. The author should rebuild the structure of this paper, it is too hard to read your paper, please highlight your contribution.

Confidence in this Review

1-Less confident (might not have understood significant parts)


Reviewer 5

Summary

The authors study a recently introduced hierarchical clustering objective function. That earlier work gave a new way to evaluate a hierarchical clustering as a whole, instead of evaluating each flat clustering in the hierarchy separately, and showed an O(log^(3/2))-approximation algorithm for the new objective. The current paper improves this approximation ratio, giving a new algorithm that achieves an O(log n) approximation. Their technique is to characterize the metric of the cost function, turn the metric into an ILP, and then round the LP relaxation. They also show their algorithm works for a generalization of the cost function, and they show hardness for achieving a constant factor approximation for the objective when using a polynomial sized LP or SDP. Finally, they show experiments comparing their algorithm to hierarchical linkage-based clustering algorithms.

Qualitative Assessment

This paper advances a nice line of work on theoretically evaluating hierarchical clustering algorithms. The strengths of this paper are as follows. They give a better approximation algorithm for the paper than the original work. Their technique, which includes characterizing the hierarchical cost function, may be of use in subsequent research on this hierarchical objective. The weaknesses are as follows. The result is a bit incremental, since they shave the approximation ratio from O(log^(3/2) n) to O(log n). Once they characterize the metric, they use standard techniques to round the LP. I am confused why they compare their algorithm to flat clusterings, when the objective is meant for high quality hierarchical clusterings. It would strengthen the paper to compare their algorithm to the original algorithm for the objective they study. Overall, this paper is good. I recommend borderline accept.

Confidence in this Review

2-Confident (read it all; understood it all reasonably well)


Reviewer 6

Summary

The paper considers the hierarchical clustering problem. The problem is as follows given set V of points with pairwise distances (the distance between i and j is denoted by k(i,j)). Find a tree T where the nodes V are leaves of the tree. The cost of two nodes are now defined as follows: k(i,j) times the total number of leaves in the subtree rooted at the lowest common ancestor of i and j in T. This objective was recently introduced in a STOC paper by Dasgupta and was shown to give attractive clusterings. In that paper Dasgupta gave a simple log(n)^(3/2) approximation algorithm that used the (complex) sparsest cut ARV approximation algorithm as a subroutine. The main result of the paper is to improve this to a log(n)-approximation algorithm. They do so by characterize the type of ultrametrics that are induced by the objective function; then they write down a clever LP relaxation that they round. In addition, they substantiate that it seems hard to achieve a constant factor approximation: any such LP/SDP cannot have polynomial size.

Qualitative Assessment

The paper is very strong technically. It contains very interesting results from a theoretical stand point. I think the introduced techniques will be of large interest to people working on clustering. The practical relevance (at this stage of the reserach) seems more limited. As it is a rather technical paper, the reading gets a little bit dense, but that is probably unavoidable. Overall, I think this is a really nice paper that I recommend to be accepted.

Confidence in this Review

2-Confident (read it all; understood it all reasonably well)